# Modular assembly of self-healing flexible thermoelectric devices with integrated cooling and heating capabilities

Xiaolong Sun[1,2,3], Yue Hou [1] ✉, Zheng Zhu[1], Bo Zhu[1], Qianfeng Ding[1], Wenjie Zhou[1], Sijia Yan[1], Zhanglong Xia[1], Yong Liu [2], Youmin Hou[4,5] ✉, Yuan Yu [6] ✉ & Ziyu Wang [1,2,3] ✉

Flexible thermoelectric devices enable direct energy conversion between heat and electrical energy, making them ideal for wearable electronics and personal thermal management. Yet, current devices lack functional module expansion, which limits the customization for diverse energy-harvesting heat sources and complicates their assembly to meet the specific power requirements of electrical appliances. Moreover, existing devices cannot be stacked to enhance thermoelectric cooling performance while maintaining flexibility and self-healing capabilities. Here, by selectively encapsulating liquid metal electrodes with carbon nanotube-doped self-healing materials with increased thermal conductivity, we substantially improve heat transfer across thermoelectric legs, thereby maximizing energy conversion efficiency. The device achieves a normalized power density of $3.14\ \mu W \cdot cm^{-2} \cdot K^{-2}$, setting a benchmark for self-healing thermoelectric devices. Benefiting from self-healing materials and liquid metal, the device demonstrates both self-healing capabilities and modular assembly, greatly expanding the application scenarios of flexible thermoelectric devices in wearable power generation and refrigeration.

The rapid advancement of wearable electronic health and exercise monitoring devices has created a demand for efficient power sources that can adapt to diverse applications[1-6]. Among the various power supply technologies[7-12], thermoelectric devices (TEDs) have received significant attention due to their ability to directly convert heat into electricity via the Seebeck effect[13,14]. As TEDs can harvest environmental thermal energy without mechanical components, they are considered promising candidates for powering wearable electronics[15-19]. Recently, the development of flexible thermoelectric devices has marked a significant step in improving both performance and conformability. Various approaches, such as utilizing organic thermoelectric materials[20,21], embedding thermoelectric fibers[22,23] or bulks[15-17,24] into elastic polymers, and printable thermoelectric inks[25,26] have shown promise in creating flexible TEDs capable of adapting to different shapes and applications.

Despite extensive efforts, several challenges impede the broader application of flexible TEDs. A primary issue is the lack of modularity and expandability of flexible TEDs, which restricts the customization for different energy-harvesting scenarios and hinders the convenient assembly for specific power requirements[27,28]. Another limitation lies in the use of polydimethylsiloxane (PDMS) as the encapsulating material in flexible TEDs. While PMDS endows inorganic TEDs with enhanced

[1]The Institute of Technological Sciences, Wuhan University, Wuhan 430072, China. [2]Key Laboratory of Artificial Micro-structures of Ministry of Education, School of Physics and Technology, Wuhan University, Wuhan 430072, China. [3]School of Physics and Microelectronics, Key Laboratory of Materials Physics of Ministry of Education, Zhengzhou University, Zhengzhou 450001, China. [4]School of Power and Mechanical Engineering, Wuhan University, 430072 Wuhan, China. [5]Max Planck Institute for Polymer Research, Ackermannweg 10, 55128 Mainz, Germany. [6]Institute of Physics (IA), RWTH Aachen University, Sommerfeldstraße 14, 52074 Aachen, Germany. ✉e-mail: yuehou@whu.edu.cn; houyoumin@whu.edu.cn; yu@physik.rwth-aachen.de; zywang@whu.edu.cn

flexibility, it also leads to high thermal resistance, which decreases the temperature difference between the two ends of the thermoelectric legs and compromises the thermoelectric efficiency of devices. Furthermore, the lack of self-healing capabilities in many flexible TED designs poses durability concerns, particularly in wearable applications where mechanical stresses are prevalent.

In this work, we developed a self-healing and modularized flexible TED by integrating the $Bi_2Te_3$-based TE legs, liquid metal EGaIn interconnects, and disulfide crosslinked polyurethane (DSPU) encapsulation. To improve the output power of this flexible TED, we created a selective encapsulation strategy to minimize the parasitic heat loss of the thermoelectric elements. The liquid metal interconnects were selectively encapsulated by using carbon nanotube-doped DSPU (CD). As a result, this thermoelectric device (CD-TED) demonstrates a high normalized power density of 3.14 µW cm$^{-2}$ K$^{-2}$. This value outperforms recently reported self-healing flexible thermoelectric generators and is among the highest levels compared with other flexible TEDs[15–17,19,25,28–33]. The good performance in conjunction with the flexibility and self-healing ability of the CD-TED greatly advances its application in wearable power generation and low-power electronics.

As our CD-TED presents exceptional self-healing properties coupled with the innovative design of LEGO-like shapes and wire circuits, it enables users to modularly assemble and expand the TEDs in both flat and vertical directions. This modular design allows customization

based on the heat source or power consumption of the electronics. Moreover, implementing a multi-layered tower-like structure enables optimal cooling for thermoelectric devices, achieving a maximum cooling temperature of 6.2 K at a current level of 0.8 A. In comparison with single-layer thermoelectric devices, this technology has led to a larger cooling temperature difference by a factor of 1.6.

## Results

### Design and fabrication of the CD-TED

Previously reported TEDs typically utilize welded metal electrodes to interconnect the TE legs, resulting in limited mechanical flexibility and high electrical resistance[16]. To address these limitations and develop a high-performance flexible TED, here we embed the TE legs into the DSPU and connect them with liquid metal electrodes, forming an electrically series and thermally parallel structure (Fig. 1a). By using liquid metal electrodes, the resistance of the TED is only 3.18 Ω. More discussions about the influence of electrical and thermal contact resistance on the thermoelectric performance of the device are provided in Supplementary Information Note 1. Note that we developed a selective encapsulation strategy to reduce the local thermal resistance of DSPU on the electrodes, thereby substantially increasing the temperature difference across the TE legs by 185% and enhancing the TE performance by 171% compared to the device without selective encapsulation.

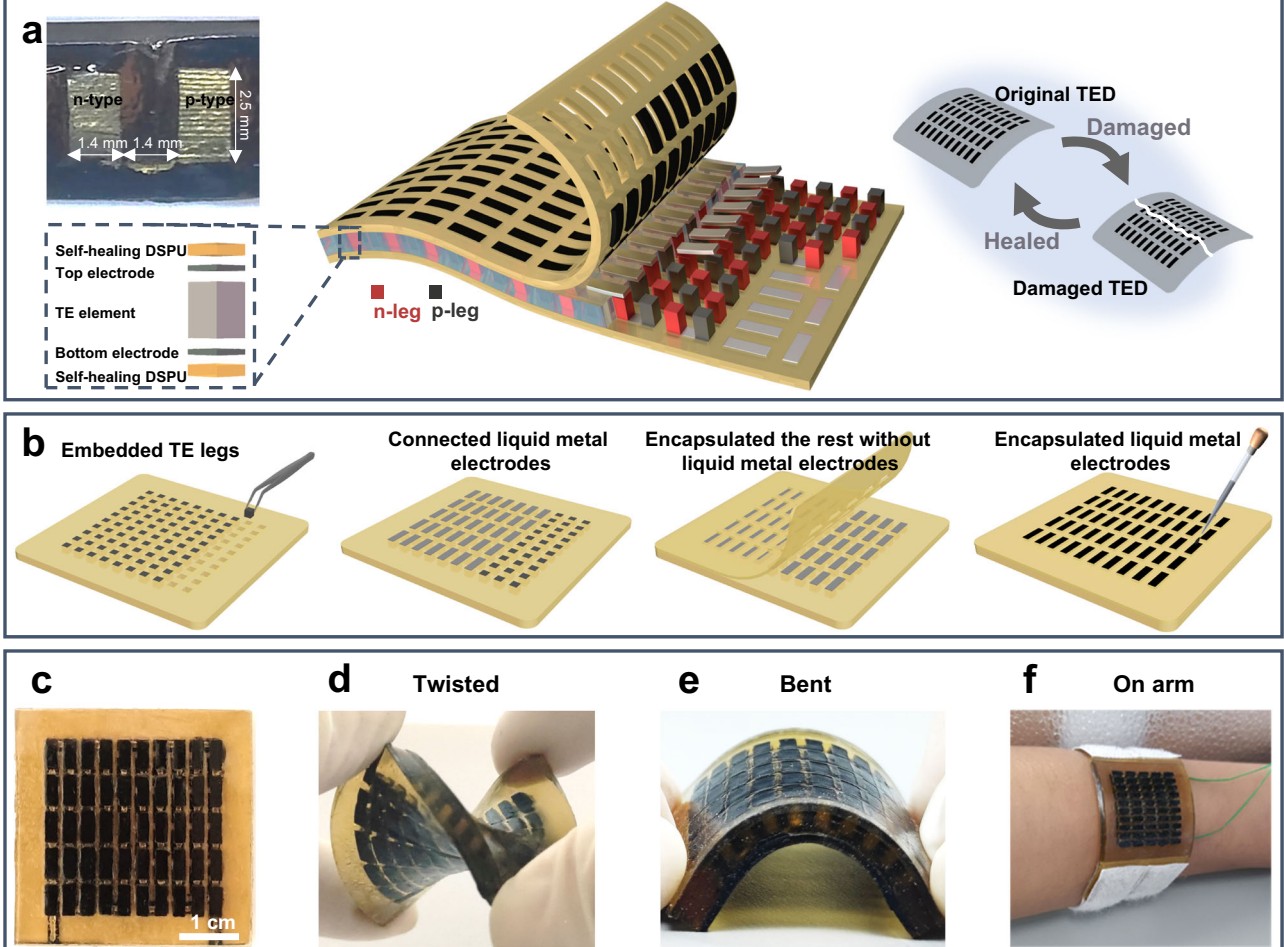

**Fig. 1 | Design and fabrication of the TED. a** Conceptual illustration of a self-healing TED with EGaIn electrodes and self-healing heat conductors (CD). The left inset is an optical image of a cross-section of the TED, and the right inset is a self-healing capability of the TED. **b** Schematic illustration of the fabrication process for the CD-TED. Optical images of the CD-TED when it is flat (**c**), twisted (**d**), bent (**e**), and worn on the arm (**f**).

We have also optimized the encapsulation layer thickness (from 0.2 mm to 1.2 mm with a step of 0.2 mm) of thermoelectric devices by combining finite element simulations with experimental validations, as shown in Supplementary Fig. 1. Both theoretical and experimental results demonstrate that a thinner encapsulation layer can enhance the effective temperature difference and output voltage across the thermoelectric legs. However, in practical applications, optimizing the performance of thermoelectric devices necessitates considering not only their thermoelectric efficiency but also their self-healing properties and fabrication feasibility. For example, after repeated bending tests, the device with a 0.8-mm encapsulation layer exhibited liquid metal leakage on its surface, while the device with a 1-mm encapsulation layer remained intact (Supplementary Fig. 2). After a comprehensive evaluation, we selected an encapsulation layer thickness of 1 mm for subsequent experiments to achieve an optimal balance between performance and practical application requirements.

Additionally, the impact of the fill factor on the output power of thermoelectric devices was investigated by finite element simulations (Supplementary Fig. 3). The results show that the output power reached its optimum at a fill factor of 26%. This value has also been utilized in the design of our thermoelectric devices (details about the geometry of the devices can be found in Supplementary Fig. 4).

As shown in Fig. 1b, the fabrication process of CD-TED began by pouring pure DSPU into a mold to create an insulating center layer. The p/n TE legs were then alternately embedded and connected using liquid metal electrodes (see Methods and Supplementary Fig. 5). Subsequently, the liquid metal electrodes were encapsulated with carbon nanotube (the thermal conductivity of CNT ranges from 1000 to 4000 W/m K)[34,35] doped DSPU with high thermal conductivity (as denoted by the dark areas in Fig. 1b), while the rest is encapsulated with DSPU to maintain the large impedance of heat transfer. Although these devices were manually assembled in this work to verify the effect of encapsulation technology on performance, automated manufacturing techniques such as using an automated dispensing machine or pick-and-place machine[29] have been demonstrated feasible, which can be employed in future research to promote a large-scale application of this technology.

Our developed CD-TED also shows an impressive self-healing capability due to the distinctive characteristics of DSPU used in encapsulation. Therefore, when the fracture surfaces of a damaged CD-TED are brought into contact, the separate parts immediately reunite owing to the DSPU and fluidity of the liquid metal (as shown in the right inset of Fig. 1a). The healed thermoelectric device has restored its mechanical and electrical functions, which will be elaborated in the following sections.

Figure 1c presents the as-prepared CD-TED, which selectively encapsulates liquid metal electrodes with carbon nanotube-doped self-healing materials. Owing to its flexibility, our CD-TED exhibits excellent compliance with mechanical deformations, such as twisting, bending, and conforming to an arm (Fig. 1d–f), highlighting its adaptability to irregular surfaces.

## TE performance of CD-TED in various conditions

To demonstrate the potential of CD-TED for powering portable electronics, we tested the voltage output and power generation performance of a CD-TED with 50 pairs of p/n TE legs. The experiments were conducted under temperature differences ($\Delta T$) ranging from 5 to 30 K using a laboratory setup (see details in Supplementary Fig. 6). Figure 2a shows the open-circuit voltage ($V_{OC}$) and output power ($P_{out}$) as a function of current ($I$) at different $\Delta T$. The CD-TED achieves a maximum $V_{OC}$ of 258 mV and $P_{out}$ of 5.55 mW, which can supply power for wearable electronics such as hearing aids and health monitors[36].

We further examined the effect of CD on heat transfer across the TE legs using numerical simulations. In the numerical model, the thermal conductivities of normal DSPU, CD, liquid metal, and TE leg at

room temperature are 0.1 W m$^{-1}$ K$^{-1}$, 0.9 W m$^{-1}$ K$^{-1}$, 25 W m$^{-1}$ K$^{-1}$, and 1.3 W m$^{-1}$ K$^{-1}$, respectively, as measured experimentally (Supplementary Figs. 7 and 8). The temperature difference applied between the top and bottom surfaces of TED was stabilized at 30 K. Figure 2b shows the simulation results of the temperature distribution for the TEDs with and without selective encapsulation of CD. For clarity, we define D-TED as the TED completely encapsulated by the pure DSPU, without carbon nanotube doping. In D-TED, the normal DSPU layers significantly hinder the heat transfer, resulting in a temperature difference of only 8.24 K across the TE leg and liquid metal, as indicated by the yellow curve. In contrast, CD-TED shows a $\Delta T = 15.21$ K across the TE legs, which is 1.85 times higher than the case of D-TED, as the doped carbon nanotube effectively reduces the heat dissipation in the DSPU layers[29]. In addition, a homogeneous distribution of CNT inside the whole DSPU contrarily decreases the effective temperature difference across the thermoelectric legs due to parasitic heat conduction, as illustrated in Supplementary Fig. 9. This further proves the effectiveness of the selective encapsulation strategy.

The large temperature difference across the TE legs leads to a better TE performance of CD-TED than the conventional design. Figure 2c compares the $V_{OC}$ and power density between D-TED and CD-TED. The D-TED generates a maximum $V_{OC}$ of 156.5 mV with a power density of 1041 $\mu$W/cm$^2$ at $\Delta T = 30$ K. By contrast, the selective encapsulation in CD-TED increased the output significantly, showing 64% higher $V_{OC}$ and 171% higher power density compared to D-TED. Figure 2d and Supplementary Table 1 compare the normalized power density and normalized Seebeck voltage of our CD-TED with other state-of-the-art flexible TEDs[15-17,19,25,28-32,37,38]. The superior TE performance of CD-TED can be attributed to the improved thermal conductivity that leads to a larger temperature difference across the TE legs at a fixed heating and cooling environment by the selective encapsulation of the carbon nanotube-doped DSPU.

In addition, the CD-TED shows stable output performance at low temperatures and humid environments. As shown in Fig. 2e, we keep $\Delta T$ of the CD-TED at ~4 K in a freezing environment and test the TE performance. Despite the decreasing ambient temperature, our device maintains the $V_{OC}$ at ~14 mV even when the temperature drops to 223.15 K (i.e., −50 °C). Meanwhile, when the air humidity in the testing chamber varies from 45% to 95% (Fig. 2f), the $V_{OC}$ of CD-TED remained nearly unchanged as the encapsulation layers of DSPU demonstrate a good hydrophobicity (see the contact angle measurements in Supplementary Fig. 10). These results suggest that CD-TED is well-suited for power generation across a variety of application scenarios, including low-temperature and high-humidity environments.

## Mechanical stability and thermal reliability of CD-TED

For wearable devices, mechanical stability and thermal reliability are of great importance in long-term operation. To evaluate the mechanical robustness of CD-TED, we conducted a cyclic bending test, with a bending radius ($r$) of 1.5 cm in both x and y directions. As shown in Fig. 3a, b, we measured the ratio of voltage ($V$) to the initial voltage ($V_0$) and the ratio of power ($P$) to the initial power ($P_0$) under 2000 bending cycles, the $V/V_0$ and $P/P_0$ at a $\Delta T$ of 30 K remain stable even after 2000 bending cycles. We also measured the ratio of resistance change ($\Delta R$) to the initial resistance ($R_0$) under 2000 bending cycles. The $\Delta R/R_0$ of the CD-TED stays below 5% when the $r$ reaches 1.5 cm (see details in Supplementary Fig. 11).

To demonstrate the bending stiffness and stretchability of the thermoelectric device, we fabricated a rectangular thermoelectric device containing 32 pairs of p/n thermoelectric legs, as shown in Supplementary Fig. 12. The bending stiffness measurements show that the CD-TED could be bent at a radius of 3.8 mm with an applied force of only 2.53 N, which could be attributed to the ultralow elastic modulus of 0.2 MPa for DSPU determined by the stress–strain curve.

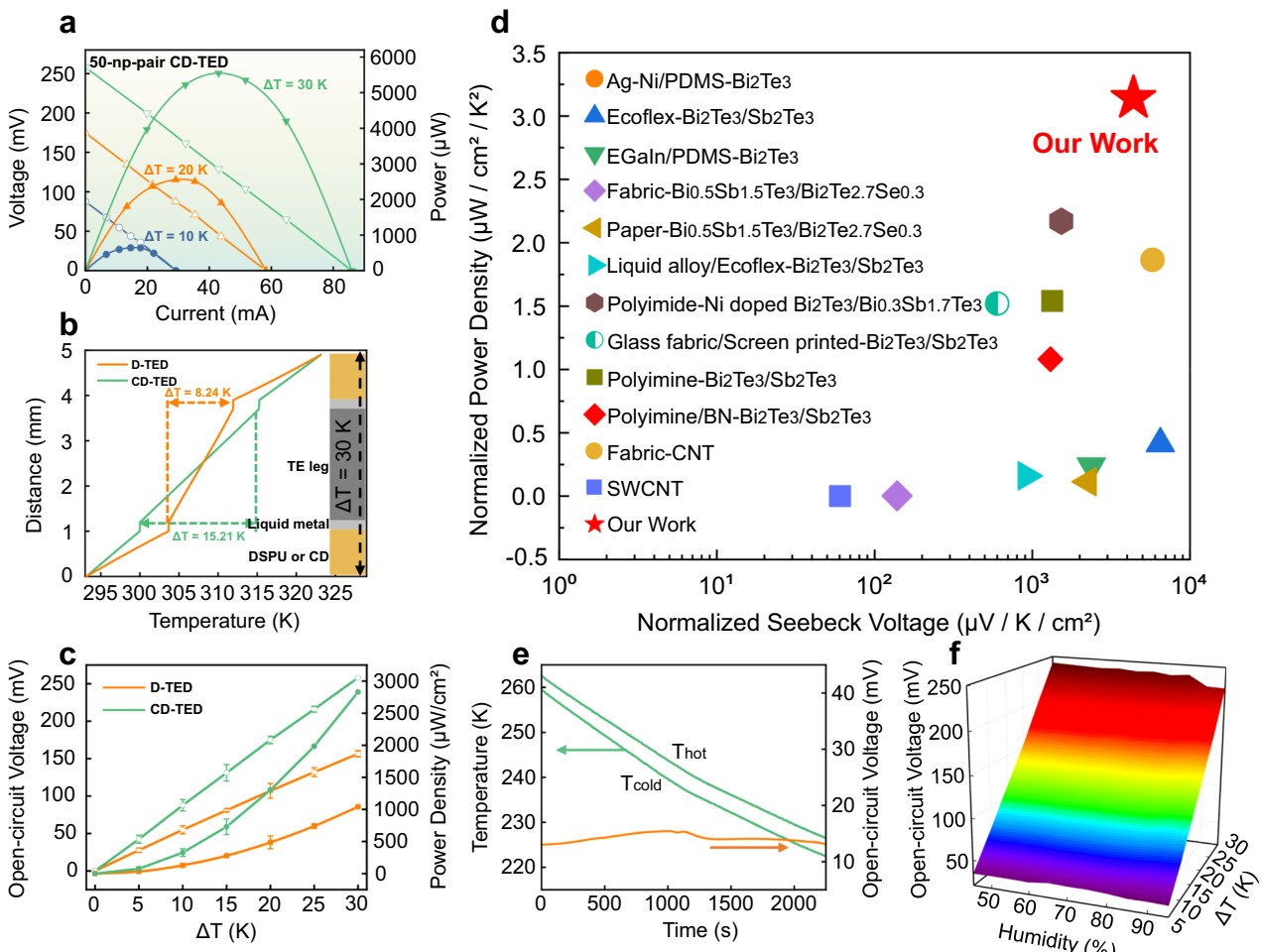

**Fig. 2 | Effect of selective encapsulation of carbon nanotube-doped DSPU (CD) on the TE performance of the self-healing TED. a** Experimentally measured TE performance of the 50-np-pair CD-TED including voltage and power as a function of current. **b** Finite element analysis (FEA) results showing the temperature distribution on the cross-section of the D-TED and CD-TED with a temperature difference of 10 K. **c** Temperature difference versus open-circuit voltage and power density of TEDs of different encapsulation methods (the error bar represents the standard deviation, reflecting the degree of data dispersion in three independent experiments). **d** Comparison of normalized power density (normalized power density is obtained by dividing the output power of a TE device by the area and the square of the temperature difference) versus normalized Seebeck voltage (the normalized Seebeck voltage is obtained by dividing the open circuit voltage of the TE device by the area and temperature difference) between this work and other typical flexible TEDs. **e** Open-circuit voltage of CD-TED under a continuously decreasing temperature condition. **f** Open-circuit voltage of CD-TED under different humidity and temperature differences.

Additionally, the resistance of the device at different bending radii remained almost unchanged. These results indicate that the thermoelectric device has excellent bending compliance, conforming to various curved structures, such as human skin, while maintaining excellent thermoelectric performance.

Moreover, we also tested the stretchability of the device. Supplementary Fig. 13 shows that the device could maintain stability and functionality under a maximum strain of 25%. No significant degradation in performance is observed after stretching at a 10% strain for 2000 cycles. This demonstrates the excellent stretchability of our thermoelectric device.

Figure 3c presents the results of the thermal reliability test. The output power and internal resistance exhibit consistent stability over 160 minutes. During the test, the hot side was maintained at a constant temperature of 323.15 K, while the cold side was exposed to natural convection in a typical indoor environment. The results not only prove the reliability and durability of thermoelectric devices in long-term usage but also indicate that they have a broad application prospect in wearable electronic devices. The stable power output can secure continuous operation of the devices in scenarios that require continuous power supply or temperature-difference energy conversion,

such as smartwatches, health-monitoring devices, or other portable electronic devices.

The excellent mechanical stability and thermal reliability of CD-TED make it suitable for wearable energy-harvesting applications. Figure 3d and Supplementary Fig. 14 show the performance of wearable CD-TED attached to the human arm. The prepared CD-TED generates an average open-circuit voltage of 7.5 and 13.5 mV when the wearer is sitting and working, respectively. This highlights the potential of CD-TED as a self-sufficient power source, capable of harnessing body heat to energize low-power wearable electronic devices at the microwatt level[36]. However, the equilibrium temperature difference across the thermoelectric legs is only about 1 K under natural convection. There are primarily two ways to increase the temperature difference, thereby enhancing the output power and voltage of our device. Finite element simulation analysis indicates that output performance improves with an increase in the height of the thermoelectric legs (Supplementary Fig. 15). In this work, we opted for a leg height of just 2.5 mm to maintain bendability. The second factor is the filling material between the legs; simulations show that using air instead of DSPU can increase the output voltage by 14.5% (Supplementary Fig. 16). Indeed, a well-considered thermal design involving an

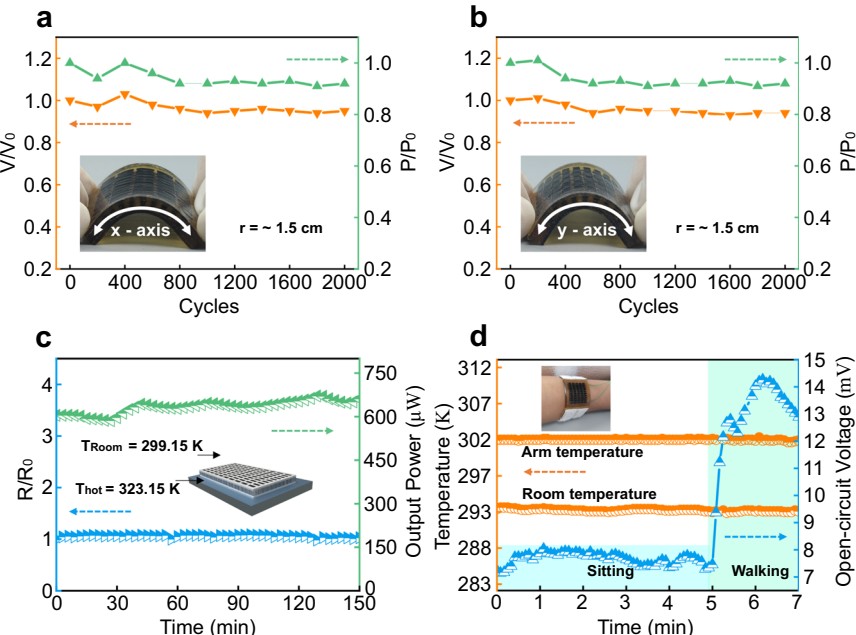

**Fig. 3 | Mechanical and thermal reliability of the CD-TED.** Experimentally measured TE performance of CD-TED after different bending cycles with different bending directions of the *x*-axis (**a**) and *y*-axis (**b**). **c** Endurance test with the hot-side temperature ($T_{hot}$) kept at 323.15 K. The cold side was natural convection, and the room temperature ($T_{room}$) was around 299.15 K. **d** The open-circuit voltage of the CD-TED worn on the arm.

optimal leg height and the use of air filling has led to improved device performance[27]. This insight will guide our future research as we strive to maximize output performance while preserving self-healing capability and flexibility.

## Self-healing property of the CD-TED

In this CD-TED, the fluidity of the liquid metal electrodes coupled with the recombination of bonds within the DSPU network confers the device with remarkable self-healing characteristics. The self-healing mechanism of DSPU is shown in Supplementary Note 2. The DSPU matrix comprises both covalent bonds (disulfide bonds) and hydrogen bonds[39,40]. The former provides dynamic crosslinking in the polymer chain, while the latter further promotes the self-healing ability. Meanwhile, we also tested the self-healing properties of the DSPU material. The DSPU elastomer was cut in half and manually recombined at room temperature with different healing times. Supplementary Fig. 17a shows the tensile stress–strain curves of the original and self-healed samples at different healing times. After 2 h, the performance curve of DSPU almost coincides with that of the original material. These results indicate that the hydrogen bonds have good self-healing efficiency within a short time at room temperature. Further self-healing is mainly dominated by the cohesive forces mediated by the insertion of IP-based segments into the aromatic disulfide metathesis. Moreover, after storing for two weeks, we retested the tensile stress-strain curves and self-healing efficiency of DSPU (Supplementary Fig. 17b). Results show that the self-healing efficiency of the material after healing for 120 min did not significantly decrease over time. This self-healing ability and mechanical properties guarantee its application in flexible devices.

Figure 4a demonstrates the self-healing performance of the CD-TED. A printed circuit board, integrating a step-up voltage converter and a light-emitting diode (LED), is successfully powered by the CD-TED (Fig. 4a and Supplementary Fig. 18). When the CD-TED is positioned on a 40 °C heating plate, it generates approximately 135 mV and the output voltage of the step-up voltage converter is approximately 3.59 V, which is adequate to activate the LED.

When the liquid metal electrodes and DSPU are cut off by a cutting blade, the LED is turned off. Upon reassembling the severed thermoelectric devices and applying gentle pressure to the damaged area, the separated liquid metal flows and reconnects. After one hour at ambient temperature, a robust network of covalent and hydrogen bonds forms at the interface, resulting in the CD-TED self-healing and the LED re-illuminating. The self-healed CD-TED exhibits comparable flexibility to its original device, enduring 2000 bending cycles without any significant alteration to its electrical resistance (Fig. 4b). It is noteworthy that the output voltages of the CD-TED and the step-up voltage converter remain consistent before and after the self-healing process (Fig. 4c). We attached the CD-TED to the convex surface of a cup filled with hot water (40 °C). Under these thermal conditions, the CD-TED successfully powers the LED via a step-up voltage converter, as depicted in Fig. 4d.

## The module assembly with CD-TED blocks

The CD-TED not only has self-healing capabilities but also has modular assembly potential, facilitated by the integration of self-healing materials and liquid metal electrodes. Figure 5a illustrates the modular assembly process, where nine individual CD-TEDs are module-assembled into a new CD-TED. The modular assembly process began with cutting off the terminal of the device to expose the liquid metal electrodes. Subsequently, these exposed terminals of the TEDs were carefully brought into physical contact. The newly assembled CD-TED emerged as a fully functional entity, demonstrating impressive flexibility and adaptability (Fig. 5a and Supplementary Fig. 19).

As shown in Fig. 5b, the open-circuit voltage of the CD-TED increases linearly as more modules are assembled. It demonstrates that the modular assembly approach is feasible and maintains performance integrity. This feature allows different users to personalize CD-TED by employing series or parallel module arrangements, depending on specific thermal conditions and the desired output characteristics. This customization is designed to attain the optimal shape factor, structural configuration, output voltage, and power output.

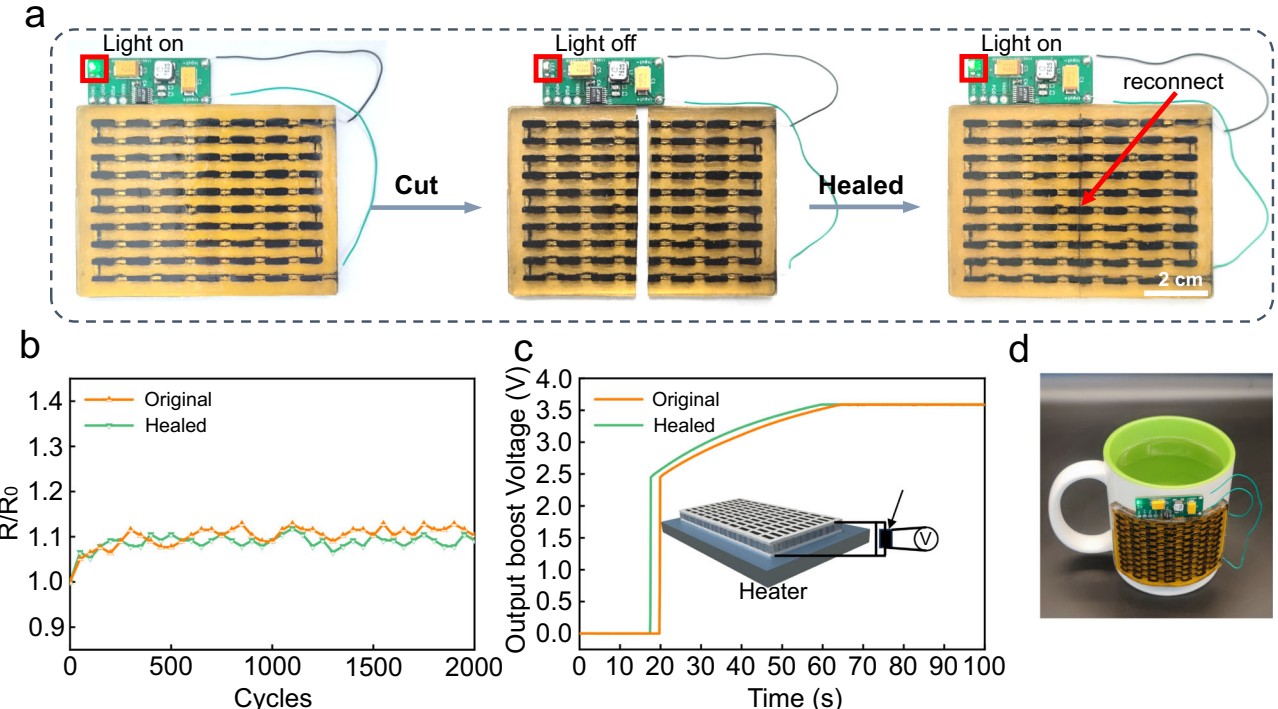

**Fig. 4 | Self-healing property of the CD-TED. a** Optical image of the TED in a self-healing test. Directly lighting a light-emitting diode (LED) by harvesting heat energy with a step-up voltage converter (before and after heating). **b** Bending cycle test of the CD-TED demonstrates remarkable stability in conductivity, both before and after the self-healing process, with no significant degradation observed even after 2000 bending cycles. **c** Output voltages of the step-up voltage converter when the TED (before and after healed) were put on a hot plate. **d** Optical image of healed TED attached to a cup.

Leveraging the Peltier effect, we applied this CD-TED for cooling applications. As depicted in Fig. 5c, a tower-like structure was implemented to enhance the TED's cooling capacity. CD-TEDs with one or two TE layers have been successfully assembled. Supplementary Fig. 20 illustrates that the CD-TED with two TE layers retains commendable bending performance. Nonetheless, it exhibits a slight decrease in flexibility when compared to its single TE layer counterpart. Upon elevating the TED to three or more layers, the increased thickness adversely affects its flexibility, rendering it unsuitable for wearable applications. Consequently, the optimal choice in this context is to utilize a two-layer TED configuration.

When a direct current is introduced into the thermoelectric device, the temperature disparity between the device's cold end and the surrounding environment escalates in tandem with the rising current. As illustrated in Fig. 5d, upon reaching a current of 0.8 A, the temperature difference between the cold end of the thermoelectric device and the ambient environment peaks. Specifically, when the thermoelectric device is configured with a two-layer tower structure, this maximum temperature difference is recorded at 6.2 K. Additionally, we have integrated a modular assembly approach with a tower structure to augment the flexibility of the thermoelectric devices. As demonstrated in Fig. 5e, both the single-layer and double-layer thermoelectric devices exhibit excellent flexibility, capable of accommodating a variety of bending curvatures to meet diverse requirements.

## Discussion

This study introduces an innovative thermoelectric device (CD-TED) that is not only flexible and self-healing but also modularly assembled, offering exceptional performance. The selective encapsulation of liquid metal electrodes with carbon nanotube-doped self-healing materials enhances the heat transfer to the TE legs while maintaining its self-healing properties and exhibiting a thermal conductivity of

$0.9\ \mathrm{W\ m^{-1}\ K^{-1}}$. The CD-TED maximizes the $\Delta T_{TE}/\Delta T_{Applied}$ ratio by up to 50%, leading to a remarkable 171% increase in power output compared to the D-TED. The finite element analysis further validates the improved heat transfer ability and TE performance, attributable to the incorporation of carbon nanotube-doped self-healing materials. Enhanced by self-healing materials and liquid metals, this thermoelectric device exhibits both self-healing capabilities and modular assembly, while also adapting to various cylindrical curvatures. These characteristics broaden the versatility of flexible thermoelectric devices, opening the avenues for wider applications in wearable energy harvesting and temperature regulation.

## Methods

### Materials

$Bi_{0.5}Sb_{1.5}Te_3$ (p-type) and $Bi_2Te_{2.7}Se_{0.3}$ (n-type) TE legs were purchased from Hubei Sagreon New Energy Technology Company, Ltd., and the size of the cuboids was custom-made with the size of 1.4 mm × 1.4 mm × 2.5 mm, the thermoelectric properties of the material can be found in the supplementary Table 2 and Supplementary Fig. 8. Poly(propylene glycol) (PPG, Mn = 4000), ditin butyl dilaurate (DBTDL, 95%), isophorone diisocyanate (mixture of isomers) (IPDI, 99%), 4-aminophenyl disulfide (APDS, 98%) were purchased from Aladdin Chemistry Co. Ltd. (Shanghai, Chain). Tetrahydrofuran (THF) was purchased directly from Sinopharm Chemical Reagent Co., Ltd. (Shanghai, China). The above materials are used as received without further purification.

### Synthesis of self-healing DSPU elastomers

The self-healing polymer was prepared according to refs. 39–41. Firstly, for PPG-IPDI prepolymer preparation, 18 mL of PPG was heated at 70 °C under an argon gas atmosphere and then reacted with IPDI (3.2 mL) for 45 min in the presence of DBTDL (100 μL). Afterward, 3.1 g of PPG-IPDI prepolymer, 1.35 mL of APDS (0.28 g), and THF mixed

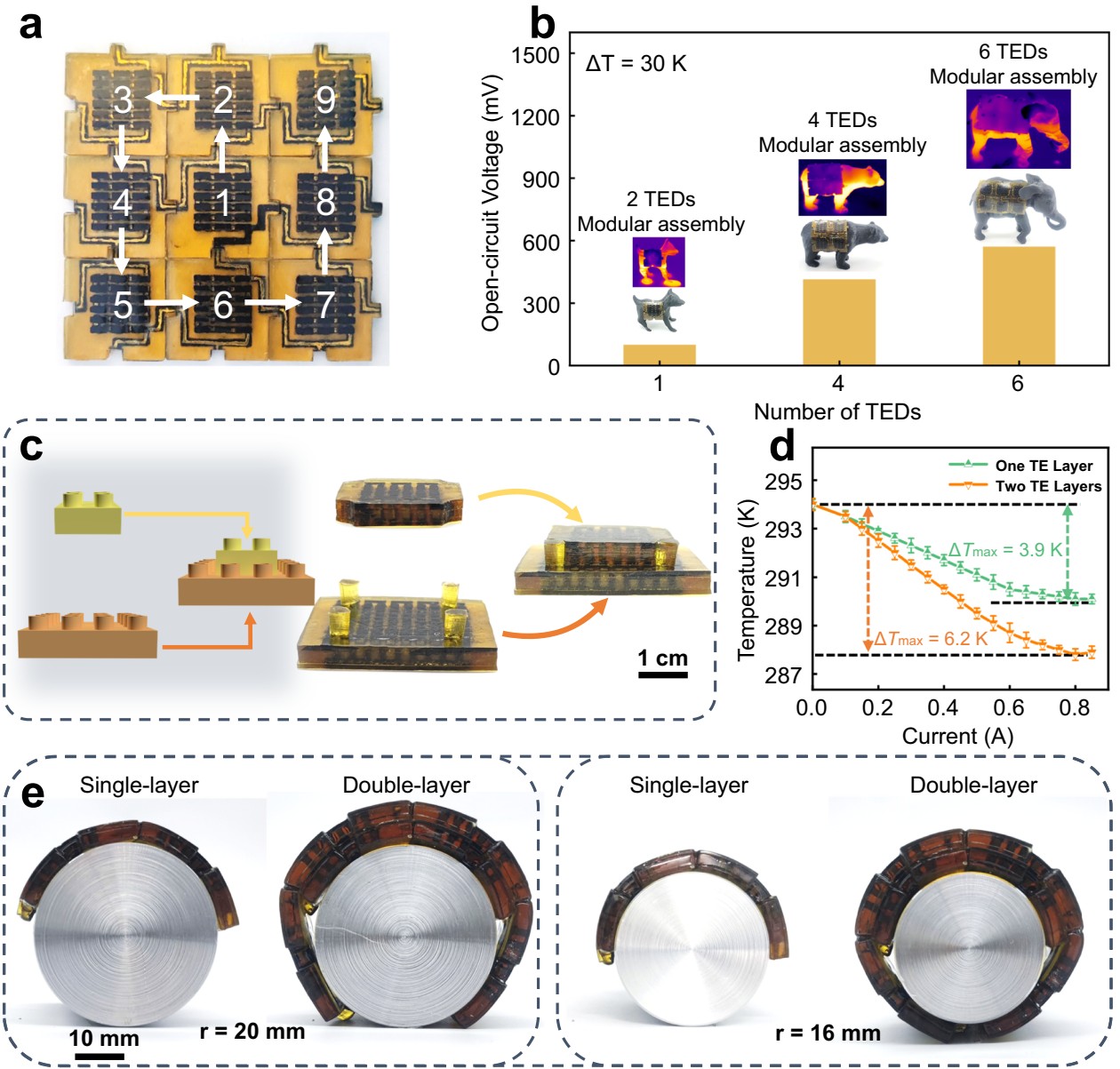

**Fig. 5 | The module-assembled CD-TED. a** Demonstration of the module-assembled TEDs with different numbers. **b** Open circuit voltage of TEDs with varying numbers of modular assemblies. **c** Optical image of one TE Layer and two TE Layers. **d** Cooling capacity of various TE Layers under different currents (the error bar represents the standard deviation, reflecting the degree of data dispersion in three independent experiments). **e** Optical image of thermoelectric devices affixed to cylindrical surfaces with varying degrees of curvature.

solution were mixed with a magnetic stirrer for 5 min, then degassed for 15 min in a vacuum chamber. Finally, the viscous reactants were transferred into the square Teflon molds and cured for 16 h at 75 °C to obtain DSPU elastomers. Self-healing material for encapsulating upper and lower electrodes was obtained by incorporating 10% carbon nanotubes into DSPU. The maximum temperature tolerance of self-healing DSPU is usually between 120 °C to 140 °C.

### Fabrication of the CD-TED

We first poured the prepared DSPU into the mold and prepared an insulating DSPU center layer for fixing the TE legs. Then, the p/n TE legs were alternately embedded in the center layer of the DSPU, and the TE legs were interconnected with LM electrodes. Subsequently, the top and bottom electrodes were covered with a layer of DSPU, leaving only the LM electrodes exposed to air, and then the LM was encapsulated with CD, and a flexible and self-healing CD-TED was finally produced.

### Characterization

The output performances of the TED were measured by a Keithley 2400 source meter. The internal resistance of the TED was measured using an electrochemical analyzer (CHI650e, CH Instruments, Inc.). The infrared image was captured by an infrared camera (Ti480 PRO FLUKE). The temperature difference between the thermoelectric devices was measured by the thermocouple (JK-8A). The homemade heater and cooler were used for heating and cooling, respectively. The thermal conductivity was measured by using a transient hot wire thermal conductivity meter (TC3000E).

### Data availability

All data necessary to understand and assess this manuscript are shown in the main text and the Supporting Information. The data that support the findings of this study are available from the corresponding author upon reasonable request.

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

## Acknowledgements

This work is supported by the National Key R&D Program of China (Grant No. 2023YFB4603800) awarded to Z.W., the National Natural Science Foundation of China (Grant No. 12302220) awarded to Y.H., and the National Natural Science Foundation of China (Grant No. 12474093) awarded to Z.W.

## Author contributions

Y.H., Y.M.H., Y.Y., and Z.W. conceived and designed experiments. X.S. prepared the thermoelectric device, processed the mechanical test, healable test, performed simulations by FEM, and conducted experiments, data collection, and analysis. Z.Z., B.Z., and Q.D. performed

structure design. W.Z. performed simulations by FEM, S.Y., and Z.X. conducted and collected healable test data. Y.L. designed circuits for devices. Y.M.H., Y.Y., and Z.W. supervised the work. X.S., Y.H., Y.Y., and Z.W. wrote the paper. All authors discussed and commented on the paper.

## Funding

## Competing interests
The authors declare no competing interests.
