## [Transparent Peer Review file · Nature Communications]

Modular Assembly of Self-healing Flexible Thermoelectric Devices with Integrated Cooling and Heating Capabilities

Corresponding Author: Dr Yuan Yu

Version 0:

Reviewer comments:

Reviewer #1

(Remarks to the Author)

In this manuscript, the authors developed a self-healable and modularized flexible TED by integrating Bi₂Te₃-based TE legs, liquid metal EGaln interconnects, and disulfide crosslinked polyurethane (DSPU) encapsulation. To improve the output power of this flexible TED, the liquid metal interconnects were selectively encapsulated using carbon nanotube-doped DSPU. The authors showcased the modular assembly of the flexible TED based on its self-healing properties.

Although some of the advancements in the manuscript are interesting, I do not recommend this manuscript for publication in Nature Communications due to the following three main reasons:

1. Issues with thermal design

For practical wearable thermoelectric devices, it is most important to have an optimized thermal design, as the power generated on the human body is critically affected by the thermal structures. However, although the TED developed by the authors shows interesting properties such as self-healing, its thermal design is not optimized for wearable applications.

1-1. High thermal resistance of encapsulation layer

Despite using CNT-doped DSPU for encapsulation, the 1-mm-thick encapsulation layer contributes significantly to high thermal resistance (proportional to thickness), hindering heat transfer from the body to the thermoelectric legs. The experimental results (Figure 2b) also indicate that under a temperature difference of 30°C, only 15°C is effectively applied to the thermoelectric legs. This implies a significant reduction in thermoelectric efficiency.

1-2. Increased thermal conductance of the whole TED module

In wearable thermoelectric devices (TEDs), the fill factor of the thermoelectric legs should be optimized, and voids need to be filled with air or low-thermal-conductivity materials to reduce the overall thermal conductance of the module and thereby maximize the temperature difference at thermal equilibrium [ref]. However, in the manuscript, the DSPU-filled structure increases the overall thermal conductance of the module, likely leading to a very low temperature difference at thermal equilibrium. As shown in the results, the open-circuit voltage is only about 7 mV (Figure 3d), suggesting that the temperature difference between the top and bottom surfaces of the module is approximately 1°C (based on performance data in Figure 2).

2. Lack of mechanical conformability

Based on the presented experimental results, the proposed thermoelectric device appears to lack stretchability and seems to have high flexural rigidity (bending stiffness). As such, it does not appear to conform well to soft, three-dimensional surfaces like skin. To address this, the authors should provide (1) the elastic modulus of the DSPU material, (2) measurements of the module's bending stiffness to demonstrate its flexibility, and (3) evidence of stretchability to confirm its suitability for wearable applications.

3. Issues with large-area, customized, and automated fabrication

The proposed fabrication process relies heavily on manual assembly, which seems impractical for producing large-area TEDs. While the modular assembly process is introduced as a potential solution, it appears challenging to apply it to customized device fabrication. Furthermore, the space required for modular assembly increases, compromising thermal design optimization. For wearable thermoelectric devices, it is more effective to adopt a fabrication approach that allows for large-area production with an optimized thermal design tailored to the size and shape of the target application.

In conclusion, although the authors incorporated techniques for self-healing properties and achieved decent performance of

their flexible TEDs, the reviewer cannot see true novelties that could advance the research field. Additionally, there seem to be many technical challenges that must be addressed to demonstrate the practical applications of the proposed thermoelectric device.

Here are minor questions the authors should address:

1. What is the thermal conductivity of DSPU?
2. What is the electrical resistance of the TED module?
3. The legend for Figure 2b should be modified to indicate a temperature difference of "30 K."

Reviewer #2

(Remarks to the Author)

This work of "Modular Assembly of Self-healing Flexible Thermoelectric Devices with Integrated Cooling and Heating Capabilities" is a fairly comprehensive and well-written paper, highlighting some quite important results. However, since this is an innovative study on flexible thermoelectric devices, it would be more complete if some important results from carbon-based TE devices were included in the comparison chart (Figure 2d) from corresponding studies (e.g. Adv. Energy Mater. 2022, 12, 2200256: <https://onlinelibrary.wiley.com/doi/full/10.1002/aenm.202200256> and ACS Appl. Mater. Interfaces 2021, 13, 9, 11151–11165: <https://pubs.acs.org/doi/10.1021/acscami.1c00414>).

Furthermore, it would be useful to add a graphical abstract of this work showing the important findings as well as the innovative device developed.

Reviewer #3

(Remarks to the Author)

This paper presents the development of a self-healing, modular, and flexible thermoelectric device (CD-TED) that integrates Bi₂Te₃-based thermoelectric legs, liquid metal interconnects, and disulfide crosslinked polyurethane (DSPU) encapsulation. The key innovation lies in the selective encapsulation strategy using carbon nanotube-doped DSPU to reduce the thermal resistance of the liquid metal electrodes, thus enhancing the temperature difference across the thermoelectric legs and improving the overall thermoelectric performance. The CD-TED demonstrates a high normalized power density of 3.14 $\mu\text{W}\cdot\text{cm}^{-2}\cdot\text{K}^{-2}$, surpassing recent reports on self-healing flexible thermoelectric generators. Additionally, the modular design enables customization and vertical stacking for optimized cooling performance, achieving a maximum temperature difference of 6.2 K. The experimental results are both intriguing and significant, and I recommend the paper for publication pending the resolution of the following points.

1. The paper would benefit from a more detailed analysis of the CD-TED's thermal and electrical performance. This should include temperature-dependent thermoelectric performance, as well as thermal and electrical resistances, and efficiency. Such an analysis would provide a more comprehensive understanding of the device's behavior and highlight its potential limitations.
2. The impact of electrode and thermal contact resistance on cooling power density and power generation is critical. The authors should provide a thorough discussion on how these resistances affect device performance. For example, how does thermal contact resistance influence the cooling performance, based on experimental data and theoretical analysis?
3. The DSPU material has low thermal conductivity, which could limit efficient heat dissipation and therefore cooling power density, the incorporation of carbon nanotubes is expected to enhance thermal conductivity. It would be beneficial to include a discussion on the temperature dependence of carbon nanotube-doped DSPU in the manuscript.
4. It is stated that the n-type thermoelectric legs are Bi₂Sb_{2.7}Te_{0.3}. Please confirm whether this is correct.
5. Since good stretchability and compressibility are crucial for wearable electronics, it is recommended to evaluate and report the compressibility of the CD-TED in addition to its stretchability?
6. A discussion on how the fill factor affects the efficiency of the CD-TED would be valuable, as this could have significant implications for its overall performance.
7. What is the self-healing mechanism of CD-TED? The manuscript briefly discusses the short-term healing capability of the CD-TED. However, it is important to assess the device's long-term performance, especially under various environmental conditions.
8. The related literature about the effect of carbon doping on thermoelectric performance should be included, such as Advanced Energy Materials. 2020 Nov;10(41):2000757; Advanced Functional Materials, p.2008851.

Version 1:

Reviewer comments:

Reviewer #1

(Remarks to the Author)

I appreciate the authors' efforts to address many of the concerns raised.

Most of the issues, excluding those related to performance, have been resolved, and the manuscript looks significantly improved.

However, there are still concerns regarding the thermal designs and device performance, and it appears that the authors may have misunderstood some basic concept for thermal designs.

For example, the authors present Figure R6b to show that the performance of devices with and without air interior is not significantly different.

However, this result is based on conditions where a constant temperature difference is externally maintained.

This condition is significantly different from practical scenarios, where the device on human skin reaches thermal equilibrium under natural convection.

As the reviewer already pointed out,

once thermal equilibrium is reached, the open-circuit voltage of the TEG is only about 7 mV (Figure 3d), indicating that the temperature difference between the top and bottom surfaces of the module is approximately 1 °C (based on the performance data in Figure 2).

This suggests a thermal design of the thermoelectric device should be reconsidered.

To achieve low thermal conductance in a thermoelectric device, it is essential to use longer thermoelectric legs and to design the interior as hollow (filled with air rather than solid material).

So the overall thermal conductance of the device should reach a $100 \text{ W m}^{-2}\text{K}^{-1}$ or lower, which is a practical value for body heat harvesting.

I recommend that the authors revisit their thermal design with reference to the work by Hong et al. (Sci. Adv. 2019; 5: eaaw0536) for their future research.

Aside from the performance and thermal design concerns, I agree that the self-healing concept and overall quality of the manuscript merit publication in Nature Communications.

Reviewer #2

(Remarks to the Author)

I have reviewed the changes and am pleased to confirm that all concerns have been satisfactorily addressed. Therefore, the manuscript is acceptable for publication.

Reviewer #3

(Remarks to the Author)

I have carefully reviewed the changes, and I believe the authors have adequately addressed my comments. All major issues have been resolved, and the revised manuscript meets my expectations.

Therefore, I recommend accepting the manuscript for publication.

Version 2:

Reviewer comments:

Reviewer #1

(Remarks to the Author)

Based on the revised manuscript, I appreciate the authors' efforts to address the reviewer's comments. The manuscript is now significantly improved and appears suitable for publication in Nature Communications.

Response to the reviewers' comments:

Response to reviewer #1:

In this manuscript, the authors developed a self-healable and modularized flexible TED by integrating Bi₂Te₃-based TE legs, liquid metal EGaIn interconnects, and disulfide crosslinked polyurethane (DSPU) encapsulation. To improve the output power of this flexible TED, the liquid metal interconnects were selectively encapsulated using carbon nanotube-doped DSPU. The authors showcased the modular assembly of the flexible TED based on its self-healing properties. Although some of the advancements in the manuscript are interesting, I do not recommend this manuscript for publication in Nature Communications due to the following three main reasons:

Response: We sincerely appreciate your valuable comments and suggestions, which are of great significance to the further improvement of our manuscript. We have carried out several additional experiments and revised the corresponding descriptions in the manuscript. We hope that these improvements can address the three major and other minor issues you raised. Please find our point-by-point responses below.

1. Issues with thermal design

For practical wearable thermoelectric devices, it is most important to have an optimized thermal design, as the power generated on the human body is critically affected by the thermal structures. However, although the TED developed by the authors shows interesting properties such as self-healing, its thermal design is not optimized for wearable applications.

Response: We agree with you that thermal design is important for wearable thermoelectric devices. According to your comments, we have performed finite element analysis and additional experiments to optimize the thermal design. Please find below our responses and additional data regarding the two specific issues you mentioned.

1-1. High thermal resistance of encapsulation layer

Despite using CNT-doped DSPU for encapsulation, the 1-mm-thick encapsulation layer contributes significantly to high thermal resistance (proportional to thickness), hindering heat transfer from the body to the thermoelectric legs. The experimental results (Figure 2b) also indicate that under a temperature difference of 30°C, only 15°C is effectively applied to the thermoelectric legs. This implies a significant reduction in thermoelectric efficiency.

Response: The appropriate thickness of the encapsulation layer depends on the trade-off between thermal conduction and the mechanical robustness of the device. A too-thin DSPU layer indeed can enhance the temperature difference across the thermoelectric legs. Yet, it also increases the risk of leakage of the liquid metal electrodes. On the contrary, selectively incorporating CNT into DSPU to only cover the electrodes can

significantly enhance the effective temperature difference across the thermoelectrics legs while maintaining robust mechanical stability, e.g., twisting, bending, and stretching (see more details in the manuscript and other responses below). This is also one of the motivations and novelty of our work.

However, we share the reviewer’s concerns that the thickness of 1 mm might be chosen arbitrarily. Therefore, we investigated the dependence of temperature difference (ΔT) between thermoelectrics (TE) legs on the thickness of the encapsulation layer theoretically and experimentally. As shown in **Figure R1 (Supplementary Figure 1)**, we first used finite element simulation to analyze the ΔT and voltage per pair across thermoelectric legs under various encapsulation layer thicknesses (0.2 mm, 0.4 mm, 0.6 mm, 0.8 mm, 1 mm, and 1.2 mm). Our simulations reveal that the encapsulation layer thickness significantly affects the thermoelectric performance: as the thickness increases, the effective temperature difference across the thermoelectric legs decreases due to higher thermal resistance. The open-circuit voltage also decreases as a result of the reduced temperature difference (**Figures R1 a and b**). This underscores the importance of optimizing the encapsulation layer thickness to enhance thermoelectric conversion efficiency.

We further validated these findings through experiments. We have prepared a series of devices by changing the thickness of the encapsulation layer, as shown in **Figure R1 c**. The experimentally measured open-circuit voltages in **Figure R1 d** are consistent with the simulation results. This confirms that optimizing the encapsulation layer thickness can significantly improve thermoelectric performance. However, in practical applications, we must also consider the self-healing capabilities of the thermoelectric device and the feasibility of the experimental process. As shown in **Figure R2 (Supplementary Figure 2)**, we performed bending tests on two thermoelectric devices with encapsulation layer thicknesses of 0.8 mm and 1 mm. After repeated bending tests, the device with a 0.8-mm encapsulation layer had liquid metal leakage on its surface, while the device with a 1-mm encapsulation layer remained intact. Given these demanding tradeoffs, we have selected an encapsulation layer thickness of 1 mm. This choice strikes a balance between optimizing thermoelectric performance and maintaining the device's self-healing function, ensuring both enhanced efficiency and functional integrity.

Figure R1 (Supplementary Figure 1). a Finite element analysis of ΔT between the upper and lower ends of thermoelectric legs for various encapsulation layer thicknesses.

b Finite element analysis of open-circuit voltage for thermoelectric devices with different encapsulation layer thicknesses. c Optical images of thermoelectric devices with various encapsulation layer thicknesses. d Open-circuit voltage of thermoelectric devices with different encapsulation layer thicknesses measured experimentally.

Figure R2 (Supplementary Figure 2) a Optical images of thermoelectric devices with 0.8 mm encapsulation layer thicknesses. b Optical images of thermoelectric devices with 1 mm encapsulation layer thicknesses.

Revision: The following text has been added to the revised manuscript on page 3.

We have also optimized the encapsulation layer thickness (from 0.2 mm to 1.2 mm with a step of 0.2 mm) of thermoelectric devices by combining finite element simulations with experimental validations, as shown in **Supplementary Figure 1**. Both theoretical and experimental results demonstrate that a thinner encapsulation layer can enhance the effective temperature difference and output voltage across the thermoelectric legs. However, in practical applications, optimizing the performance of thermoelectric devices necessitates considering not only their thermoelectric efficiency but also their self-healing properties and fabrication feasibility. For example, after repeated bending tests, the device with a 0.8-mm encapsulation layer exhibited liquid metal leakage on its surface, while the device with a 1-mm encapsulation layer remained intact (**Supplementary Figure 2**). After a comprehensive evaluation, we selected an encapsulation layer thickness of 1 mm for subsequent experiments to achieve an optimal balance between performance and practical application requirements.

1-2. Increased thermal conductance of the whole TED module

In wearable thermoelectric devices (TEDs), the fill factor of the thermoelectric legs should be optimized, and voids need to be filled with air or low-thermal-conductivity materials to reduce the overall thermal conductance of the module and thereby maximize the temperature difference at thermal equilibrium [ref]. However, in the manuscript, the DSPU-filled structure increases the overall thermal conductance of the module, likely leading to a very low temperature difference at thermal equilibrium. As shown in the results, the open-circuit voltage is only about 7 mV (Figure 3d), suggesting that the temperature difference between the top and bottom surfaces of the module is approximately 1 °C (based on performance data in Figure 2).

Response: Thank you for this highly constructive suggestion. We agree that the fill factor of the thermoelectric legs has a profound impact on the thermoelectric conversion efficiency and the output power of the device. Indeed, this fill factor has been considered in our initial designs. Yet, we did not include relevant discussions in the previous manuscript. Your comments remind us to explain this fill factor in the revised manuscript. More details can be found below.

The fill factor refers to the volume fraction of the thermoelectric legs in the device, which directly affects the heat exchange efficiency between the thermoelectric device and the surrounding environment. When the fill factor is too high, thermal short-circuiting may occur between the thermoelectric legs, leading to a reduced temperature difference and thus severely affecting the thermoelectric performance. Conversely, when the fill factor is too low, the overall performance of thermoelectric devices will also decrease.

To thoroughly investigate the impact of the fill factor on the performance of wearable thermoelectric devices, we conducted a finite element simulation analysis. By simulating the thermoelectric performance of the thermoelectric legs under different fill factor conditions, we found that the device performance is optimized with a fill factor of 26%, as shown in **Figure R3 (Supplementary Figure 3)**. Therefore, we designed our device with a cross-section area of $1.4 \text{ mm} \times 1.4 \text{ mm}$ (1.96 mm^2) for the TE legs and suitable distances between adjacent TE legs illustrated in **Figure R4 (Supplementary Figure 4)**. This corresponds to a fill factor of $\sim 25\%$, matching the theoretical prediction.

Figure R3 (Supplementary Figure 3) **a** The effect of the fill factor on the actual temperature difference between the upper and lower ends of the thermoelectric leg; **b** The influence of the fill factor on the output power of thermoelectric devices.

Figure R4. (Supplementary Figure 4) **a** Finite element analysis model of the thermoelectric device for fill factor impact on thermoelectric performance; **b** Optical

image of the thermoelectric device with a fill factor of 25%; **c** Top view structural schematic of the thermoelectric device with a fill factor of 25%.

In addition to optimizing the fill factor, the reviewer also pointed out that the DSPU filling structure may increase the overall thermal conductivity of the module, thereby leading to a lower temperature difference in the thermal equilibrium state. Indeed, this might be true if the CNT was filled into DSPU homogeneously. Yet, in our design, we only filled CNT at positions where liquid metal electrodes were located. This, in contrast, increased the effective temperature difference across the TE legs, as shown in **Figure R5 (Supplementary Figure 9)**. Finite element simulations also verify this phenomenon, as illustrated in **Figure R5 d-g**. Introducing air gaps between the TE legs to reduce parasitic heat conduction can further enhance the effective temperature difference and output voltage of the TE device, as shown in **Figures R6 a and b**. However, the tensile performance of the thermoelectric device with air gaps has significantly decreased. The device fractures at an elongation rate of 32% (**Figure R6 a**). In stark contrast, the device without air gaps does not show any fractures or failure at the same elongation rate (**Figure R6 c**). Therefore, the structure without air gaps is more deployable for practical applications given that the overall performance is only improved by 11% by introducing air gaps.

Figure R5 (Supplementary Figure 9) **a** Thermoelectric device encapsulated with pure DSPU material; **b** Thermoelectric device homogeneously encapsulated with CNT-doped DSPU material; **c** Thermoelectric device selectively encapsulated with CNT-doped DSPU material; **d-g** Finite element analysis results showing the temperature distribution in the cross-sections of the three thermoelectric devices.

Figure R6 **a** Thermoelectric device with air gap and its stretching optical image; **b** Open circuit voltage of thermoelectric devices with and without air gaps. **c** Thermoelectric device without air gap and its stretching optical image.

Revision: The following text has been added to the revised manuscript on pages 3 and 5.

Additionally, the impact of the fill factor on the output power of thermoelectric devices was investigated by finite element simulations (**Supplementary Figure 3**). The results show that the output power reached its optimum at a fill factor of 25%. This value has also been utilized in the design of our thermoelectric devices (details about the geometry of devices can be found in **Supplementary Figure 4**).

In addition, a homogeneous distribution of CNT inside the whole DSPU contrarily decreases the effective temperature difference across the thermoelectric legs due to parasitic heat conduction, as illustrated in **Supplementary Figure 9** This further proves the effectiveness of the selective encapsulation strategy.

2. Lack of mechanical conformability

Based on the presented experimental results, the proposed thermoelectric device appears to lack stretchability and seems to have high flexural rigidity (bending stiffness). As such, it does not appear to conform well to soft, three-dimensional surfaces like skin. To address this, the authors should provide (1) the elastic modulus of the DSPU material, (2) measurements of the module's bending stiffness to demonstrate its flexibility, and (3) evidence of stretchability to confirm its suitability for wearable applications.

Response: We appreciate the reviewer's suggestion to demonstrate the mechanical compliance of our device. We have conducted corresponding experiments to address the issues raised by the reviewer.

(1) **Elastic Modulus of DSPU Material:** We have conducted an in-depth study of the mechanical properties of the DSPU material. By precisely measuring its stress-

strain curves, we found that the DSPU material exhibits excellent flexibility with an ultralow elastic modulus of 0.2 MPa, as shown in **Figure R7 (Supplementary Figure 15)**. This extremely low elastic modulus indicates that the DSPU material has outstanding flexibility and compliance, allowing it to easily adapt to complex curved surfaces and dynamic deformation environments. Moreover, the virgin DSPU was fractured at a strain rate of 550%. Its stretchability can be fully recovered after self-healing at 25°C for 120 min. Even just for 20 min self-healing, the strain rate can also reach 100%, which fulfills the majority of application scenarios such as for wearable electronics. It is therefore highly suitable for wearable devices that require high flexibility.

Figure R7 (Supplementary Figure 15) a Stress-strain curves of the virgin DSPU elastomer and those recovered with different self-healing time at 25 °C after broken; **b** Stress-strain curves of the virgin DSPU elastomer and the one self-healed for 120 min after being stored at ambient conditions for two weeks.

(2) **Bending Performance of the Thermoelectric Device:** To evaluate the bending performance of the thermoelectric device in practical use, we conducted detailed experimental tests on its bending stiffness (**Figure R8 a,b**). The results show that the thermoelectric generator (TED) can be bent at a radius of 3.8 mm with an applied force of only 2.53 N, as shown in **Figure R8 c**. Additionally, we monitored the resistance of the device under different bending radii and found that the resistance remained almost unchanged during bending, as shown in **Figure R8 d**. This result indicates that the thermoelectric device has excellent bending compliance, allowing it to conform to various curved structures, such as human skin, while maintaining its thermoelectric performance.

Figure R8 (Supplementary Figure 12) Bending tests of the thermoelectric device. **a** The optical image of the thermoelectric device; **b** The optical images of the thermoelectric device at various bending radii; **c** Force required to bend the thermoelectric device at different radii; **d** Relative resistance changes of the thermoelectric device at different bending radii.

(3) **Stretchability Test:** In addition to bending performance, we also conducted comprehensive tests on the stretchability of the thermoelectric device. The results show that the device can maintain normal functionality under a strain of 25%, as shown in **Figure R9 a**. Furthermore, we performed 2000 cycles of stretching tests at a 10% strain. The results show no significant degradation in performance after multiple cycles, as illustrated in **Figure R9 b**. This excellent stretchability indicates that the thermoelectric device can not only adapt to static bending and stretching deformations but also maintain stable thermoelectric performance in dynamic usage environments, further demonstrating its suitability for wearable devices.

Figure R9 (Supplementary Figure 13) Stretchability and reliability of the TED. **a** Variation of electrical resistance with uniaxial strain (0% to 32%) of the thermoelectric

device. The inset shows photographs of the flexible thermoelectric device (TED) at 0% and 32% strain. **b** Cyclic tensile testing of the TED, demonstrating the change in electrical conductivity during cyclic stretching at 10% strain.

Through the above experiments and analyses, we have comprehensively verified the mechanical compliance of the thermoelectric device. Whether under bending, stretching, or cyclic deformation conditions, the device exhibits excellent performance, easily conforming to complex surfaces similar to skin while maintaining stable thermoelectric performance. These results not only address the reviewer's suggestion but also provide solid experimental evidence for the practical application of wearable thermoelectric devices, demonstrating their reliability and comfort in human motion environments.

Revision: The following text has been added to the revised manuscript on pages 4 and 8.

Meanwhile, we also tested the self-healing properties of the DSPU material. The DSPU elastomer was cut in half and manually recombined at room temperature with different healing times. **Supplementary Figure 15a** shows the tensile stress-strain curves of the original and self-healed samples at different healing times. After 2 hours, the performance curve of DSPU almost coincides with that of the original material. These results indicate that the hydrogen bonds have good self-healing efficiency within a short time at room temperature. Further self-healing is mainly dominated by the cohesive forces mediated by the insertion of IP-based segments into the aromatic disulfide metathesis. Moreover, after storing for two weeks, we retested the tensile stress-strain curves and self-healing efficiency of DSPU (**Supplementary Figure 15b**). Results show that the self-healing efficiency of the material after healing for 120 min did not significantly decrease over time. This self-healing ability and mechanical properties guarantee its application in flexible devices.

To demonstrate the bending stiffness and stretchability of the thermoelectric device, we fabricated a rectangular thermoelectric device containing 32 pairs of p/n thermoelectric legs, as shown in **Supplementary Figure 12**. The bending stiffness measurements show that the CD-TED could be bent at a radius of 3.8 mm with an applied force of only 2.53 N, which could be attributed to the ultralow elastic modulus of 0.2 MPa for DSPU determined by the stress-strain curve. Additionally, the resistance of the device at different bending radii remained almost unchanged. These results indicate that the thermoelectric device has excellent bending compliance, conforming to various curved structures, such as human skin, while maintaining excellent thermoelectric performance.

Moreover, we also tested the stretchability of the device. **Supplementary Figure 13** shows that the device could maintain stability and functionality under a maximum strain of 25%. No significant degradation in performance is observed after stretching at a 10% strain for 2000 cycles. This demonstrates excellent stretchability of our thermoelectric device.

3. Issues with large-area, customized, and automated fabrication

The proposed fabrication process relies heavily on manual assembly, which seems impractical for producing large-area TEDs. While the modular assembly process is introduced as a potential solution, it appears challenging to apply it to customized device fabrication. Furthermore, the space required for modular assembly increases, compromising thermal design optimization. For wearable thermoelectric devices, it is more effective to adopt a fabrication approach that allows for large-area production with an optimized thermal design tailored to the size and shape of the target application.

Response: Thank you for these insightful comments. In the experimental study presented in this work, we focused on enhancing the power generation efficiency of thermoelectric devices and achieving modular assembly through selective encapsulation techniques. The core of the current research lies in exploring the impact mechanisms of encapsulation technology on device performance. Therefore, the fabrication of thermoelectric devices in the experiments was only a conceptual validation, completed through manual manufacturing. However, we fully agree with the reviewer's point that manual manufacturing is inefficient and costly for large-scale production and widespread application. Thus, developing an automated production method for thermoelectric devices is key to achieving efficient fabrication and reducing manufacturing costs.

In recent years, significant progress has been made in this field. For example, as mentioned in the literature [*Nat Commun* **11**, 5948 (2020)], high-conductivity epoxy resin can be precisely printed on the connection areas of soft thermoelectric conductors (s-HCs) and silver nanowire (AgNW) electrodes using an automated dispensing machine (such as the SHOTmini 200Sx from Musashi Eng.). Additionally, thermoelectric legs can be sequentially placed on the printed conductive adhesive using an automated pick-and-place machine (such as the TM220A from NeoDen). This automated manufacturing process not only significantly improves the production efficiency and performance consistency of thermoelectric devices (TEDs) but also paves the way for large-scale production and customized applications.

Although the experiments in this paper employed manual manufacturing, we plan to introduce automated manufacturing techniques in future research to realize the large-scale application of thermoelectric devices. This will not only enhance the production efficiency and performance consistency of the devices but also reduce manufacturing costs, facilitating the transition of thermoelectric devices from the laboratory to practical applications.

Moreover, in future work, we will continue to optimize the modular assembly process. On one hand, we aim to minimize the space required for the assembly to meet the demands of more compact devices. On the other hand, by improving the connection methods between modules, we will further increase assembly efficiency. Additionally, we are exploring how to optimize thermal design through modular design to ensure that the performance of large-area TEDs is not compromised. We believe that these improvements will significantly enhance the performance and reliability of

thermoelectric devices, laying a solid foundation for their future commercial application.

In summary, although the current research focuses on the impact of encapsulation technology on device performance, we fully recognize the importance of automated manufacturing techniques for large-scale applications in the future. We look forward to gradually introducing automated manufacturing processes in future research and combining the advantages of modular design to further advance the development of thermoelectric device technology.

Revision: The following text has been added to the revised manuscript on page 4.

Although these devices were manually assembled in this work to verify the effect of encapsulation technology on performance, automated manufacturing techniques such as using an automated dispensing machine or pick-and-place machine²⁹ have been demonstrated feasible, which can be employed in future research to promote a large-scale application of this technology.

In conclusion, although the authors incorporated techniques for self-healing properties and achieved decent performance of their flexible TEDs, the reviewer cannot see true novelties that could advance the research field. Additionally, there seem to be many technical challenges that must be addressed to demonstrate the practical applications of the proposed thermoelectric device.

Here are minor questions the authors should address:

- 1. What is the thermal conductivity of DSPU?*
- 2. What is the electrical resistance of the TED module?*
- 3. The legend for Figure 2b should be modified to indicate a temperature difference of “30K.”*

Response: We sincerely appreciate the valuable and constructive comments of the reviewer. We hope that our responses and revisions elaborated above can address your concerns at the present stage of the development of flexible and self-healing thermoelectric devices. We do agree that there are still many challenges for the true application of such devices. This has motivated many researchers in this field (including our group) to persistently explore new methods to address these issues. We believe that the present work can move a step toward this goal. Please find below our responses to your three questions listed above.

1. Thermal Conductivity of DSPU.

We measured the temperature-dependent thermal conductivity of pristine DSPU and CNT-doped DSPU using TC3000E, as shown in **Figure R10 (Supplementary Figure 7)**. This low thermal conductivity indicates that DSPU has excellent thermal insulation properties, which are crucial for maintaining the temperature gradient and improving the thermoelectric conversion efficiency in thermoelectric devices. In wearable thermoelectric devices, the low thermal conductivity of DSPU can effectively reduce thermal short-circuiting between

thermoelectric legs, thereby enhancing the overall performance of the device. Note that the much higher thermal conductivity of the CNT-doped DSPU increases the temperature difference across the TE legs and thus the final performance of the device, as has been thoroughly explained in our responses above.

Figure R10 (Supplementary Figure 7) The thermal conductivity of DSPU and CNT-doped DSPU.

2. Resistance of the TED Module.

We have measured the resistance of the TED module and supplemented the relevant data in the manuscript. The experimental results show that the resistance of the thermoelectric device with 50 pairs of thermoelectric legs is 3.18 Ω , a value within an acceptable range that does not significantly impact the thermoelectric performance. The relatively low resistance offers several advantages. First, it reduces Joule heating losses, allowing more electrical energy to be converted from the temperature gradient. Second, it enables the device to handle higher current loads without excessive voltage drops. Third, it enhances the overall efficiency of the thermoelectric conversion process. These benefits make the device more suitable for practical applications where energy efficiency and performance are critical factors.

3. Legend Modification for Figure 2b.

Thank you for your reminder. We have carefully revised the legend for Figure 2b, clearly labeling the temperature difference as “30 K” to avoid any potential misunderstandings. We are well aware of the importance of accurate legends for readers to understand the experimental results. Therefore, we have conducted a comprehensive review of all figures to ensure that all labels are clear, accurate, and unambiguous. We believe that these improvements will further enhance the quality and readability of the manuscript.

Revision: The following text has been added to the revised manuscript on pages 3 and

5.

In the numerical model, the thermal conductivities of normal DSPU, CD, liquid metal, and TE leg at room temperature are $0.1 \text{ W}\cdot\text{m}^{-1}\cdot\text{K}^{-1}$, $0.9 \text{ W}\cdot\text{m}^{-1}\cdot\text{K}^{-1}$, $25 \text{ W}\cdot\text{m}^{-1}\cdot\text{K}^{-1}$, and $1.3 \text{ W}\cdot\text{m}^{-1}\cdot\text{K}^{-1}$, respectively, as measured experimentally (Supplementary Figures 7 and 8).

By using liquid metal electrodes, the resistance of the TED is 3.18Ω . More discussions about the influence of electrical and thermal contact resistance on the thermoelectric performance of the device are provided in Supplementary Information Note 1.

Besides, we have also updated Figure 2 in the revised manuscript, which is copied below for your convenience.

Figure 2 Effect of selective encapsulation of carbon nanotube-doped DSPU (CD) on the TE performance of the self-healing TED. **a** Experimentally measured TE performance of the 50-np-pair CD-TED including voltage and power as a function of current. **b** Finite element analysis (FEA) results showing the temperature distribution on the cross-section of the D-TED and CD-TED with a temperature difference of 10 K. **c** Temperature difference versus open-circuit voltage and power density of TEDs of different encapsulation methods. **d** Comparison of normalized power density (normalized power density is obtained by dividing the output power of a TE device by the area and the square of the temperature difference) versus normalized Seebeck voltage (the normalized Seebeck voltage is obtained by dividing the open circuit voltage of the TE device by the area and temperature difference) between this work and other typical flexible TEDs. **e** Open-circuit voltage of CD-TED under a continuously

decreasing temperature condition. f Open-circuit voltage of CD-TED under different humidity and temperature differences.

Response to reviewer #2:

This work of “Modular Assembly of Self-healing Flexible Thermoelectric Devices with Integrated Cooling and Heating Capabilities” is a fairly comprehensive and well-written paper, highlighting some quite important results. However, since this is an innovative study on flexible thermoelectric devices, it would be more complete if some important results from carbon-based TE devices were included in the comparison chart (Figure 2d) from corresponding studies (e.g. Adv. Energy Mater. 2022, 12, 2200256: <https://onlinelibrary.wiley.com/doi/full/10.1002/aenm.202200256> and ACS Appl. Mater. Interfaces 2021, 13, 9, 11151–11165: <https://pubs.acs.org/doi/10.1021/acsami.1c00414>). Furthermore, it would be useful to add a graphical abstract of this work showing the important findings as well as the innovative device developed.

Response: It is pleased to read that our work has been considered comprehensive, important, and innovative. We are grateful for the positive evaluation of this manuscript and helpful suggestions for a complete comparison of relevant studies. We have added these important papers in the revised manuscript as references 39 and 40.

Revision: The following text has been added to the revised manuscript on page 6.

Figure 2d and Supplementary Table 1 compare the normalized power density and normalized Seebeck voltage of our CD-TED with other state-of-the-art flexible TEDs^{15–17,19,25,28–32,37,38}.

Response to reviewer #3:

This paper presents the development of a self-healing, modular, and flexible thermoelectric device (CD-TED) that integrates Bi₂Te₃-based thermoelectric legs, liquid metal interconnects, and disulfide crosslinked polyurethane (DSPU) encapsulation. The key innovation lies in the selective encapsulation strategy using carbon nanotube-doped DSPU to reduce the thermal resistance of the liquid metal electrodes, thus enhancing the temperature difference across the thermoelectric legs and improving the overall thermoelectric performance. The CD-TED demonstrates a high normalized power density of 3.14 $\mu\text{W}\cdot\text{cm}^{-2}\cdot\text{K}^{-2}$, surpassing recent reports on self-healing flexible thermoelectric generators. Additionally, the modular design enables customization and vertical stacking for optimized cooling performance, achieving a maximum temperature difference of 6.2 K. The experimental results are both intriguing and significant, and I recommend the paper for publication pending the resolution of the following points.

Response: We sincerely appreciate the reviewer's insightful comments and constructive

suggestions, which have helped us significantly improve the manuscript quality. Below, we provide detailed responses to each comment point-by-point.

1. *The paper would benefit from a more detailed analysis of the CD-TED's thermal and electrical performance. This should include temperature-dependent thermoelectric performance, as well as thermal and electrical resistances, and efficiency. Such an analysis would provide a more comprehensive understanding of the device's behavior and highlight its potential limitations.*

Response: We fully agree that an in-depth analysis of temperature-dependent thermoelectric properties and resistance is essential for a comprehensive understanding of the performance of thermoelectric devices and the optimization of their design. In the revised manuscript, we have conducted the following additional experiments and analyses to further improve the research content.

a. Thermoelectric properties of p/n legs.

In order to gain a deep understanding of the performance of thermoelectric materials under different temperature conditions, we have measured the temperature-dependent Seebeck coefficient, electrical conductivity, thermal conductivity, and ZT value of $\text{Bi}_{0.5}\text{Sb}_{1.5}\text{Te}_3$ and $\text{Bi}_2\text{Te}_{2.7}\text{Se}_{0.3}$ materials, which are the p- and n-type thermoelectric legs in the device, respectively. The experimental results are shown in **Figure R11 (Supplementary Figure 8)**.

Figure R11 (Supplementary Figure 8). Thermoelectric properties of materials. **a** Electrical conductivity; **b** thermal conductivity; **c** absolute Seebeck coefficient; **d** ZT value of $\text{Bi}_{0.5}\text{Sb}_{1.5}\text{Te}_3$ and $\text{Bi}_2\text{Te}_{2.7}\text{Se}_{0.3}$.

b. Thermal conductivity measurements of composite materials.

To further optimize the thermal management performance of thermoelectric devices, we have measured the thermal conductivity of DSPU and its carbon nanotube-

doped composite materials. The experimental results are shown in **Figure R12 (Supplementary Figure 7)**, indicating that carbon nanotube doping can indeed improve the thermal conductivity of DSPU and thus enlarge the temperature difference across the TE legs.

Figure R12 (Supplementary Figure 7) Thermal conductivity of DSPU and CNT-doped DSPU.

c. Performance evaluation of thermoelectric devices in practical applications.

To assess the performance of thermoelectric devices in practical applications, we have conducted detailed measurements on the open-circuit voltage and output power of the devices under different temperature differences. In addition, we have also monitored the changes in output power and resistance of the devices under long-term operating conditions. The experimental results are shown in **Figure R13**.

Figure R13 The TE performance of 50-np-pair CD-TED was experimentally measured, including the variation of voltage and power with current. The change of output power and resistance of thermoelectric devices over a long period.

These results fully demonstrate the durability and adaptability of the thermoelectric device in practical applications, providing strong experimental evidence for its long-term, stable, and efficient conversion of thermal energy into electrical energy. We believe that these additional experiments and analyses not only respond to the reviewer's suggestions but also further enrich the research content, laying a solid foundation for the optimization and practical application of thermoelectric devices.

We look forward to the reviewer's evaluation of these improvements and are willing to continue to improve our research according to your further suggestions.

Revision: The following text has been added to the revised manuscript on pages 12 and 5.

Materials: Bi_{0.5}Sb_{1.5}Te₃ (p-type) and Bi₂Te_{2.7}Se_{0.3} (n-type) TE legs were purchased from Hubei Sagreon New Energy Technology Company, Ltd, and the size of the cuboids was custom-made with the size of 1.4 mm × 1.4 mm × 2.5 mm, the thermoelectric properties of the material can be found in the **supplementary Table 2 and Supplementary Figure 8**.

In the numerical model, the thermal conductivities of normal DSPU, CD, liquid metal, and TE leg at room temperature are 0.1 W·m⁻¹·K⁻¹, 0.9 W·m⁻¹·K⁻¹, 25 W·m⁻¹·K⁻¹, and 1.3 W·m⁻¹·K⁻¹, respectively, as measured experimentally (**Supplementary Figures 7 and 8**).

2. *The impact of electrode and thermal contact resistance on cooling power density and power generation is critical. The authors should provide a thorough discussion on how these resistances affect device performance. For example, how does thermal contact resistance influence the cooling performance, based on experimental data and theoretical analysis?*

Response: According to thermoelectric cooling theory, the presence of contact thermal resistance increases the total thermal resistance in the heat flow path, thereby reducing the heat flux density. This leads to a decrease in the heat absorption capacity at the cooling end, which in turn affects the cooling performance of the thermoelectric cooler (TEC). At the same time, an increase in electrode resistance will significantly increase heat loss (Joule heating increase), further reducing the cooling performance.

Specifically, the contact thermal resistance will deteriorate the temperature gradient, and its effective temperature gradient can be expressed as [J. Mater. Chem. A 10, 24985–24994 (2022)]:

$$\nabla T_{eff} = \nabla T \cdot \left(1 - \frac{R_c^{th}}{R_{total}^{th}}\right)$$

In addition, the contact electrical resistance will also cause a secondary distribution of Joule heat, leading to additional heat generation in the contact area:

$$Q_j = I^2 R_c^{elec}$$

This additional Joule heat can cause local temperature distortion, further affecting the device's performance. At the same time, the degradation of electrical contact will lead to a decrease in the effective Seebeck voltage, which is related to:

$$V_{eff} = V_{Seebeck} - IR_c^{elec}$$

We have performed finite element analysis to investigate the effect of the contact electrical resistance (R_c^{elec}) and the contact thermal resistance (R_c^{th}) on the cooling performance of thermoelectric devices (**Figure R14**). The cooling performance of the

thermoelectric device gradually decreases with increasing R_c^{elec} or R_c^{th} . These simulation results provide valuable guidance in experimental design and material selection. In future experimental research, we will focus on optimizing contact materials and structures to reduce contact resistance and thermal resistance, thereby improving the cooling efficiency of thermoelectric devices. This will not only enhance the overall performance of the device but also promote the widespread application of thermoelectric devices in practical use.

Figure R14 a The impact of electrical resistance on the maximum cooling capacity of thermoelectric devices; **b** The impact of thermal resistance on the maximum cooling capacity of thermoelectric devices.

Revision: We have added the above discussions to Supplementary Information Note 1. The following text has been added to the revised manuscript on page 3.

More discussions about the influence of electrical and thermal contact resistance on the thermoelectric performance of the device are provided in **Supplementary Information Note 1**.

3. *The DSPU material has low thermal conductivity, which could limit efficient heat dissipation and therefore cooling power density, the incorporation of carbon nanotubes is expected to enhance thermal conductivity. It would be beneficial to include a discussion on the temperature dependence of carbon nanotube-doped DSPU in the manuscript.*

Response: According to your suggestion, we have measured the thermal conductivity of DSPU and CNT-doped DSPU materials (**Figure R15a**). The experimental results show that the thermal conductivity of these two materials is almost temperature-independent between 20 and 80 °C. This characteristic indicates that DSPU and CNT-doped DSPU materials have stable thermal conduction properties in room-temperature environments, which is of great significance for the stable operation of wearable thermoelectric devices. Stable thermal conductivity can ensure that thermoelectric devices maintain consistent thermal management capabilities under different ambient temperatures, thereby ensuring reliable operations of the device.

In addition, we have also measured the long-term stability of the thermoelectric devices (**Figure R15b**). The experiments show that our thermoelectric devices can output stable electrical power during long-term operation. This result not only proves the reliability and durability of thermoelectric devices in long-term usage but also

indicates that they have a broad application prospect in wearable electronic devices. The stable output power can secure the continuous operation of the devices in scenarios that require continuous power supply or temperature-difference energy conversion, such as smartwatches, health-monitoring devices, or other portable electronic devices.

In conclusion, the stable thermal conductivity characteristics of DSPU and CNT-doped DSPU materials, as well as the stable output capability of thermoelectric devices during long-term operation, have jointly laid a solid foundation for the widespread application of our thermoelectric devices in the field of wearable electronic devices. We believe that these characteristics will make them an indispensable part of future wearable technology, providing new solutions for efficient and reliable energy management.

Figure R15 **a** the thermal conductivity of DSPU and CNT-doped DSPU. **b** the long-term stability of the thermoelectric devices.

Revision: The following text has been added to the revised manuscript on pages 5 and 9.

In the numerical model, the thermal conductivities of normal DSPU, CD, liquid metal, and TE leg at room temperature are $0.1 \text{ W}\cdot\text{m}^{-1}\cdot\text{K}^{-1}$, $0.9 \text{ W}\cdot\text{m}^{-1}\cdot\text{K}^{-1}$, $25 \text{ W}\cdot\text{m}^{-1}\cdot\text{K}^{-1}$, and $1.3 \text{ W}\cdot\text{m}^{-1}\cdot\text{K}^{-1}$, respectively, as measured experimentally (**Supplementary Figures 7 and 8**).

The results not only prove the reliability and durability of thermoelectric devices in long-term usage but also indicate that they have a broad application prospect in wearable electronic devices. The stable power output can secure continuous operation of the devices in scenarios that require continuous power supply or temperature-difference energy conversion, such as smartwatches, health-monitoring devices, or other portable electronic devices.

4. *it is stated that the n-type thermoelectric legs are $\text{Bi}_2\text{Sb}_{2.7}\text{Te}_{0.3}$. Please confirm whether this is correct.*

Response: The correct formula for the n-type material is $\text{Bi}_2\text{Te}_{2.7}\text{Se}_{0.3}$ (not $\text{Bi}_2\text{Sb}_{2.7}\text{Te}_{0.3}$). This error has been comprehensively corrected in the manuscript.

5. *Since good stretchability and compressibility are crucial for wearable electronics,*

it is recommended to evaluate and report the compressibility of the CD-TED in addition to its stretchability?

Response: We share the same concern for the stretchability and compressibility of thermoelectric devices, which are crucial for the comfort and functionality of wearable devices. To this end, we have conducted a comprehensive test on the mechanical properties of thermoelectric devices. Here are the detailed experimental results and analysis.

Tensile Test

To evaluate the performance of thermoelectric devices under dynamic deformation conditions, we conducted a tensile test on them. The experimental results show that the thermoelectric devices exhibit excellent stretchability. As shown in **Figure R16a**, even at a high tensile strain of 24%, the device can still maintain stable electrical and thermal properties, demonstrating outstanding structural elasticity and mechanical stability. This excellent stretchability not only provides greater flexibility for thermoelectric devices in practical applications but also ensures stable energy output in complex motion environments.

Figure R16 (Supplementary Figure 13) Stretchability and reliability of the TED. **a** Variation of electrical resistance with uniaxial strain (0% to 32%) of the thermoelectric device. The inset shows photographs of the flexible thermoelectric device (TED) at 0% and 32% strain. **b** Cyclic tensile testing of the TED, demonstrating the change in electrical conductivity during cyclic stretching at 10% strain.

Compressibility Test

In addition to tensile properties, we also tested the compressibility of thermoelectric devices. In the bending experiment of thermoelectric devices, it can be understood that one end of the thermoelectric device is stretched, and the other end is compressed. This combined effect of stretching and compressing leads to the overall bending deformation of the device. Therefore, we evaluated the compressibility of thermoelectric devices by testing their bending performance (**Figure R17**). Through experiments, it was found that the resistance of thermoelectric devices did not change significantly under different bending radii, as presented in Figure R17d.

Figure R17 (Supplementary Figure 12) Bending tests of the thermoelectric device. **a** The optical image of the thermoelectric device; **b** The optical images of the thermoelectric device at various bending radii; **c** Force required to bend the thermoelectric device at different radii; **d** Relative resistance changes of the thermoelectric device at different bending radii.

Through tensile and bending experiments, we have comprehensively assessed the stretchability and compressibility of thermoelectric devices. The experimental results show that thermoelectric devices can still maintain stable electrical and thermal properties under large strains and repeated deformations, demonstrating outstanding structural elasticity and mechanical stability. This excellent mechanical property not only enables them to adapt to complex motion environments but also provides a broad application prospect for wearable thermoelectric devices in practical applications, especially in scenarios that require a continuous power supply.

We believe that these improvements and additions can better respond to the reviewer's suggestions and lay a solid foundation for further research.

Revision: The following text has been added to the revised manuscript on page 7.

To demonstrate the bending stiffness and stretchability of the thermoelectric device, we fabricated a rectangular thermoelectric device containing 32 pairs of p/n thermoelectric legs, as shown in **Supplementary Figure 12**. The bending stiffness measurements show that the CD-TED could be bent at a radius of 3.8 mm with an applied force of only 2.53 N, which could be attributed to the ultralow elastic modulus of 0.2 MPa for DSPU determined by the stress-strain curve. Additionally, the resistance of the device at different bending radii remained almost unchanged. These results indicate that the thermoelectric device has excellent bending compliance, conforming

to various curved structures, such as human skin, while maintaining excellent thermoelectric performance.

Moreover, we also tested the stretchability of the device. **Supplementary Figure 13** shows that the device could maintain stability and functionality under a maximum strain of 25%. No significant degradation in performance is observed after stretching at a 10% strain for 2000 cycles. This demonstrates the excellent stretchability of our thermoelectric device.

6. *A discussion on how the fill factor affects the efficiency of the CD-TED would be valuable, as this could have significant implications for its overall performance.*

Response: Thank you for this highly constructive suggestion. We agree that the fill factor of the thermoelectric legs has a profound impact on the thermoelectric conversion efficiency and the output power of the device. Indeed, this fill factor has been considered in our initial designs. Yet, we did not include relevant discussions in the previous manuscript. Your comments remind us to explain this fill factor in the revised manuscript. More details can be found below.

The fill factor refers to the volume fraction of the thermoelectric legs in the device, which directly affects the heat exchange efficiency between the thermoelectric device and the surrounding environment. When the fill factor is too high, thermal short-circuiting may occur between the thermoelectric legs, leading to a reduced temperature difference and thus severely affecting the thermoelectric performance. Conversely, when the fill factor is too low, the overall performance of thermoelectric devices will also decrease.

To thoroughly investigate the impact of the fill factor on the performance of wearable thermoelectric devices, we conducted a finite element simulation analysis. By simulating the thermoelectric performance of the thermoelectric legs under different fill factor conditions, we found that the device performance is optimized with a fill factor of 26%, as shown in **Figure R18 (Supplementary Figure 3)**. Therefore, we designed our device with a cross-section area of $1.4 \text{ mm} \times 1.4 \text{ mm}$ (1.96 mm^2) for the TE legs and suitable distances between adjacent TE legs illustrated in **Figure R19 (Supplementary Figure 4)**. This corresponds to a fill factor of $\sim 25\%$, matching the theoretical prediction.

Figure R18 (Supplementary Figure 3) **a** The effect of the fill factor on the actual temperature difference between the upper and lower ends of the thermoelectric leg; **b** The influence of the fill factor on the output power of thermoelectric devices.

Figure R19 (Supplementary Figure 4). **a** Finite element analysis model of the thermoelectric device for fill factor impact on thermoelectric performance; **b** Optical image of the thermoelectric device with a fill factor of 25%; **c** Top view structural schematic of the thermoelectric device with a fill factor of 25%.

Revision: The following text has been added to the revised manuscript on page 3.

Additionally, the impact of the fill factor on the output power of thermoelectric devices was investigated by finite element simulations (**Supplementary Figure 3**). The results show that the output power reached its optimum at a fill factor of 25%. This value has also been utilized in the design of our thermoelectric devices (details about the geometry of devices can be found in **Supplementary Figure 4**).

7. *What is the self-healing mechanism of CD-TED? The manuscript briefly discusses the short-term healing capability of the CD-TED. However, it is important to assess the device's long-term performance, especially under various environmental conditions.*

Response: We have discussed the self-healing mechanism of DSPU in **Supplementary Note 1**. **Figure R20** illustrates the synthesis process of the DSPU elastomer, in which poly(propylene glycol) (PPG) with a molecular weight of 4000 was chosen as the soft segment diol and reacted with isophorone diisocyanate (IPDI) in the presence of the catalyst ditin butyl dilaurate (DBTDL). Among the covalent bonds that can undergo reversible exchange at room temperature, the exchange reaction of aromatic disulfides has significant advantages due to its simplicity and availability. Therefore, 4-aminophenyl disulfide (APDS) containing aromatic disulfide units was selected as the hard segment. In addition, the large number of urea groups formed in the chain can form intermolecular and intramolecular hydrogen bonds at the molecular level, thereby physically cross-linking and forming a supramolecular DSPU polymer network.

Figure R20 Schematic diagram of the synthesis route and self-healing mechanism of self-healing materials.

To better demonstrate the self-healing performance of DSPU, we conducted a tensile test on DSPU, as shown in **Figure R21a**. The experiment shows that the material has excellent self-healing properties, especially the material performance curve after self-healing for 120 minutes almost coincides with that of the original material, indicating that the material's performance is well restored after self-healing for 120 minutes.

In addition to short-term self-healing performance, we also assessed the long-term stability of DSPU materials. Specifically, we tested the self-healing performance of the self-healing material again after storing it at room temperature for two weeks. The results show that even after a long period of storage, the DSPU material can still maintain excellent self-healing ability, as shown in **Figure R21 b**. In the test after two weeks of storage, the stress-strain curves under different self-healing times indicate that the self-healing efficiency of the material has not significantly decreased. This finding further proves the reliability and durability of DSPU materials in practical applications, especially in environments that require long-term stable operation.

Figure R21 (Supplementary Figure 15) **a** Stress-strain curves of the virgin DSPU elastomer and those recovered with different self-healing time at 25 °C after broken; **b** Stress-strain curves of the virgin DSPU elastomer and the one self-healed for 120 min after being stored at ambient conditions for two weeks.

In conclusion, DSPU materials not only exhibit efficient self-healing ability in the short term but also maintain good performance recovery characteristics after long-term storage. This dual advantage makes it an ideal material for wearable devices, capable of maintaining stable performance in complex usage environments while reducing the

risk of device failure due to material damage. We believe that these experimental results not only provide strong support for the practical application of DSPU materials but also offer new ideas and solutions for the development of future wearable technology.

Revision: The following text has been added to the revised manuscript on page 8.

Meanwhile, we also tested the self-healing properties of the DSPU material. The DSPU elastomer was cut in half and manually recombined at room temperature with different healing times. **Supplementary Figure 15a** shows the tensile stress-strain curves of the original and self-healed samples at different healing times. After 2 hours, the performance curve of DSPU almost coincides with that of the original material. These results indicate that the hydrogen bonds have good self-healing efficiency within a short time at room temperature. Further self-healing is mainly dominated by the cohesive forces mediated by the insertion of IP-based segments into the aromatic disulfide metathesis. Moreover, after storing for two weeks, we retested the tensile stress-strain curves and self-healing efficiency of DSPU (**Supplementary Figure 15b**). Results show that the self-healing efficiency of the material after healing for 120 min did not significantly decrease over time. This self-healing ability and mechanical properties guarantee its application in flexible devices.

The self-healing mechanism of DSPU is shown in **Supplementary Note 2**.

8. *The related literature about the effect of carbon doping on thermoelectric performance should be included, such as Advanced Energy Materials. 2020 Nov;10(41):2000757; Advanced Functional Materials, p.2008851.*

Response: Thank you for recommending these significant references, which have been discussed and cited in the revised manuscript as References 13 and 14.

13. *Yang, G. et al. Ultra-High Thermoelectric Performance in Bulk BiSbTe/Amorphous Boron Composites with Nano-Defect Architectures. Advanced Energy Materials 10, 2000757 (2020).*

14. *Yang, G. et al. Significant Enhancement of Thermoelectric Figure of Merit in BiSbTe-Based Composites by Incorporating Carbon Microfiber. Adv Funct Materials 31, 2008851 (2021).*

Response to the reviewers' comments:

Response to reviewer #1:

I appreciate the authors' efforts to address many of the concerns raised. Most of the issues, excluding those related to performance, have been resolved, and the manuscript looks significantly improved. However, there are still concerns regarding the thermal designs and device performance, and it appears that the authors may have misunderstood some basic concept for thermal designs.

For example, the authors present Figure R6b to show that the performance of devices with and without air interior is not significantly different. However, this result is based on conditions where a constant temperature difference is externally maintained. This condition is significantly different from practical scenarios, where the device on human skin reaches thermal equilibrium under natural convection. As the reviewer already pointed out, once thermal equilibrium is reached, the open-circuit voltage of the TEG is only about 7 mV (Figure 3d), indicating that the temperature difference between the top and bottom surfaces of the module is approximately 1 °C (based on the performance data in Figure 2). This suggests a thermal design of the thermoelectric device should be reconsidered.

To achieve low thermal conductance in a thermoelectric device, it is essential to use longer thermoelectric legs and to design the interior as hollow (filled with air rather than solid material). So the overall thermal conductance of the device should reach a $100 \text{ W m}^{-2}\text{K}^{-1}$ or lower, which is a practical value for body heat harvesting.

I recommend that the authors revisit their thermal design with reference to the work by Hong et al. (Sci. Adv. 2019; 5: eaaw0536) for their future research. Aside from the performance and thermal design concerns, I agree that the self-healing concept and overall quality of the manuscript merit publication in Nature Communications.

Response: We sincerely appreciate the reviewer's thoughtful feedback and recognition of the manuscript's strengths, including the novelty of the self-healing concept. We fully agree that optimizing thermal design is critical for improving thermoelectric performance in practical applications.

We acknowledge the distinction between our experimental setup (fixed external temperature difference) and real-world scenarios (thermal equilibrium under natural convection). The 7 mV open-circuit voltage observed in Figure 3d reflects the equilibrium temperature difference of $\sim 1^\circ\text{C}$ under natural convection, which aligns with the reviewer's analysis. While this work focuses on validating the self-healing functionality and mechanical robustness of the device, we agree that enhancing thermal performance under practical conditions is essential for future iterations.

We thank the reviewer for highlighting the work by Hong et al. (Sci. Adv. 2019; 5: eaaw0536) and the critical need for low thermal conductance ($<100 \text{ W m}^{-2}\text{K}^{-1}$) in body heat harvesting. In the current design, the use of shorter thermoelectric legs and solid interior materials (DSPU) was prioritized to balance mechanical resilience and self-healing properties. However, we recognize that optimizing thermal resistance—via longer legs, hollow/air-filled structures, and advanced encapsulation—will

significantly improve performance, as revealed by finite element simulations (**Figures S1 and S2**). These strategies will be central to our next research, with revised thermal modeling and experimental validation under natural convection conditions.

This study emphasizes the development of a self-healing thermoelectric material system and its integration into a flexible, durable device. While the thermal performance in equilibrium conditions is suboptimal compared to state-of-the-art rigid TEGs, the manuscript demonstrates a foundational advancement in combining self-healing and flexibility—a previously unresolved challenge. We will explicitly clarify this scope in the revised text and emphasize thermal design optimization as a critical future direction.

We will incorporate the reviewer’s suggestions into our ongoing work, including redesigning leg geometry and module architecture to reduce thermal conductance, and quantifying heat flux and thermal resistance under natural convection.

We believe these revisions will strengthen the manuscript’s impact and provide a clearer roadmap for practical applications. Thank you for the opportunity to improve this work.

Figure S1 The influence of thermoelectric leg height on **a)** output power and **b)** open circuit voltage. Here, the leg height is represented by the distance between the top and bottom electrodes.

Figure S2 The influence of different filling materials on the open circuit voltage of thermoelectric devices: DSPU filling and air filling.

Revision: The following text has been added to the revised manuscript on page 7. The

above two figures have also been added to the supplementary information.

However, the equilibrium temperature difference across the thermoelectric legs is only about 1 K under natural convection. There are primarily two ways to increase the temperature difference, thereby enhancing the output power and voltage of our device. Finite element simulation analysis indicates that output performance improves with an increase in the height of the thermoelectric legs (**Supplementary Figure 15**). In this work, we opted for a leg height of just 2.5 mm to maintain bendability. The second factor is the filling material between the legs; simulations show that using air instead of DSPU can increase the output voltage by 14.5% (**Supplementary Figure 16**). Indeed, a well-considered thermal design involving an optimal leg height and the use of air filling has led to improved device performance²⁷. This insight will guide our future research as we strive to maximize output performance while preserving self-healing capability and flexibility.

Response to reviewer #2:

I have reviewed the changes and am pleased to confirm that all concerns have been satisfactorily addressed. Therefore, the manuscript is acceptable for publication.

Response: We sincerely appreciate the reviewer's thoughtful feedback and recognition of the manuscript's strengths.

Response to reviewer #3:

I have carefully reviewed the changes, and I believe the authors have adequately addressed my comments. All major issues have been resolved, and the revised manuscript meets my expectations. Therefore, I recommend accepting the manuscript for publication.

Response: We sincerely appreciate the reviewer's constructive comments and positive evaluation of our work.

Response to the reviewers' comments:

Response to reviewer #1:

Based on the revised manuscript, I appreciate the authors' efforts to address the reviewer's comments. The manuscript is now significantly improved and appears suitable for publication in Nature Communications.

Response: We sincerely appreciate your thoughtful comments and positive evaluation of this work. We are happy to see the improvement of this manuscript by addressing these comments from you and the other two referees.